# SpectraLDS: Provable Distillation for Linear Dynamical Systems

**Devan Shah**[1] **Shlomo Fortgang**[1] **Sofiia Druchyna**[1]

**Elad Hazan**[1,2]

[1]Computer Science Department, Princeton University
[2]Google DeepMind Princeton

## Abstract

We present the first provable method for identifying symmetric linear dynamical systems (LDS) with accuracy guarantees that are independent of the systems' state dimension or effective memory. Our approach builds upon recent work that represents symmetric LDSs as convolutions learnable via fixed spectral transformations. We show how to invert this representation, thereby recovering an LDS model from its spectral transform and yielding an end-to-end convex optimization procedure. This distillation preserves predictive accuracy while enabling constant-time and constant-space inference per token, independent of sequence length. We evaluate our method, SpectraLDS, as a component in sequence prediction architectures and demonstrate that accuracy is preserved while inference efficiency is improved on tasks such as language modeling.

## 1 Introduction

The emergence of attention-based transformer architectures has revolutionized sequence modeling tasks, particularly in natural language processing [37] and large-scale sequence-to-sequence learning [16, 29]. These transformer models rely on the self-attention mechanism, which allows each token in a sequence to attend to every other token, enabling strong contextual understanding. However, this approach suffers from quadratic complexity in sequence length, making it computationally expensive for longer sequences. Recent research has thus explored alternative, more efficient architectures that preserve expressiveness while reducing computational costs for long sequences. Among these are attention-free approaches such as convolution-based or state-space models (SSMs) [9, 2, 22, 28], which can offer sub-quadratic or even near-linear time generation.

The most basic SSM, which is the starting point for all aforementioned models, are linear dynamical systems (LDS), a foundational framework for modeling sequential dependencies in control theory, signal processing, and learning theory [20, 13]:

$$x_t = Ax_{t-1} + Bu_t, \quad \hat{y}_t = Cx_t + Du_t. \tag{1}$$

Here $u_t$ represents the input sequence, $x_t$ encodes the past state information, and $\hat{y}_t$ approximates the target sequence. By maintaining a fixed latent state, LDSs enable efficient inference and allow for efficient memory utilization. However, gradient-based approaches for learning LDSs suffer from exploding or vanishing gradients, particularly when modeling systems with long-term memory or large hidden state dimension [14].

To address these limitations, Agarwal et al. [2] leveraged the spectral filtering method [14] and introduced the Spectral Transform Unit (STU), a convex relaxation that shifts from learning the hidden

39th Conference on Neural Information Processing Systems (NeurIPS 2025).

transition matrix $A$ directly to a reparameterization of the LDS impulse response that learns how input signals convolve with fixed spectral filters. This approach provably preserves the expressiveness of an LDS with symmetric $A$ while making the training problem far more tractable. In practice, it has proven robust for systems requiring long-range memory and empirical results show that hybrid STU models—with alternating attention and STU layers—can match or even surpass purely attention-based architectures, as well as other popular hybrid state-space models, on long-context tasks such as language modeling and time-series prediction [22]. Furthermore, unlike self-attention, which requires $O(L^2)$ operations per $L$ tokens during training and identically for inference with KV-caching, the STU operates in $O(L \log L)$ operations per $L$ tokens during training and $O(L \log^3 L)$ operations per $L$ tokens during inference through algorithms based on the Fast Fourier Transform [1, 22].

However, although STUs capture LDS-like dynamics using spectral filtering, there has been no straightforward way to convert (or "distill") a trained STU layer back into explicit LDS form. Moreover, it has remained unclear whether every STU layer can be represented as a LDS with provably small error. Such a distillation from an STU layer to an LDS layer with hidden-dimension $h$ would permit recurrent inference in place of convolution operations, reducing the cost of generating $L$ tokens during inference to $O(h \cdot L)$, while maintaining the training robustness guarantees of the STU.

Such a distillation is especially desirable in the light of the resurgence of SSMs as scalable alternatives to Transformers in long-context tasks [9, 10, 34]. Thus, our work allows one to retain the STU's performance on long sequences while enabling **logarithmic-time per-token generation**—providing an appealing alternative to both long convolutions and Transformer-based self-attention caching.

## 1.1 Our Contribution

We present a novel technique for distilling STU filters into LDS form, achieving a substantial reduction in operations—from $O(\log^3 L)$ to $O(\log L)$ per token[1] during generation—while preserving the STU's expressivity and performance. Moreover, due to the training stability guarantees of the STU architecture, even when learning a marginally stable symmetric LDS or an LDS with large hidden dimension, this distillation procedure provides the **first provable method** to directly learn the parameters of a **symmetric LDS of arbitrarily high effective memory and with bounded noise**. Specifically:

- In Algorithm 2, we show how to convert a learned STU into explicit system matrix parameters whose recurrence can be computed in logarithmic time.

- We provide a theoretical analysis of this distillation in Theorem 1, and empirically demonstrate that the new LDS form incurs negligible degradation in modeling quality while improving the autoregressive generation speed.

- We demonstrate in Section 6 that by applying SpectraLDS to trained STUs, we maintain the same accuracy and attain $O(\log L)$ per-token computational generation cost.

- As a consequence of our theoretical analysis, we show in Section 5.4 how we can efficiently convert a symmetric LDS of **arbitrary high dimension**, via an intermediate STU learning step, to one of $O(\log L)$ dimension with minimal error.

We open-source the SpectraLDS code at `https://github.com/dshah02/SpectraLDS`.

## 2 Related Work

State-space models have long been a cornerstone of control theory and signal processing. The simplest variants, linear dynamical systems, provide a succinct way to capture temporal dependencies via a hidden state that evolves with linear transitions. Classical methods include the Kalman filter [20], which remains widely used due to its robust theoretical properties and computational efficiency.

In recent years, a new wave of SSMs have emerged as efficient alternatives to attention-based methods for long-sequence tasks, promising sub-quadratic or even near-linear complexity without

---

[1]We expect the actual reduction to be much larger: from $\sqrt{L}$, which is a practical method for generation, as opposed to $\log^3 L$, which is a complex algorithm, especially to implement efficiently on a GPU.

compromising expressive power. Models like S4 [10] and its diagonal variants [11] exploit structured state matrices to learn long-range dependencies, while works such as Hyena, Mega, and Mamba [34, 24, 9] incorporate gating or convolution-based parameterizations to compete against (and sometimes outperform) transformers in language modeling, time-series analysis, and other long-context applications. Their growing popularity and the challenges of training SSMs underscore the need for methods with greater training robustness, efficiency, and performance guarantees. The Spectral Transform Unit (STU) [2] lies squarely in this tradition, offering a powerful convex relaxation for training LDS-like systems that achieves impressive empirical results on long-context tasks. Our work builds directly on this line of research, introducing the first method to distill a learned STU layer into an explicit LDS with provable guarantees, thereby unifying the convex training advantages of spectral filtering with the real-time inference benefits of a recurrent LDS.

Our contributions also align with the long tradition of system identification for LDSs, where the aim is to learn the hidden transition and output matrices $(A, B, C, D)$ from observed sequences. Early influential approaches—such as the Ho–Kalman algorithm [15], Kung's method [21], and the Eigensystem Realization Algorithm (ERA) [18] —rely on linear-algebraic decompositions (e.g., SVD) of Hankel matrices consistent with observations. Modern variants allow for single-trajectory identification [31], and subsequent refinements like MOESP and N4SID [17] added stochastic noise modeling, while prediction-error and maximum-likelihood methods improved estimation accuracy and statistical efficiency. More recent lines of work incorporate regularization and spectral methods (e.g., stable-spline kernels, sparse identification) to yield more robust or interpretable LDS representations. In the context of distillation, these methods still require matrix decompositions that scale super-linearly in the problem size.

In contrast to many previous approaches, since the STU's parameterization avoids direct reconstruction of the system matrix $A$, distillation from a STU layer into an LDS remains agnostic to hidden dimension. Moreover, as our method uses a fixed and abstract Hankel matrix—rather than having to construct it anew from observed data— we can perform a significant part of the distillation computation offline.

Finally, we highlight the recent closely related work of [28], which distills state-space-like recurrences from convolution-based sequence models. Their *Laughing Hyena* method accelerates long convolutional filters by approximating them with a diagonal LDS, thus allowing constant-time generation at inference. While this approach generalizes to any convolution-based model (e.g., Hyena [34]), it does not provide formal guarantees on distillation quality. In contrast, we focus on the STU's spectral filters, which have expressive power comparable to a symmetric LDS with real eigenvalues, and present the first theoretical framework to convert the filters into such an LDS with provable bounds on distillation quality. By leveraging the STU's fixed bank of spectral filters, our method preserves long-sequence expressiveness while achieving a symmetric LDS realization with a guaranteed approximation error (see Section 5).

## 3 Token Generation and Complexity for Language Modeling

In this section, we summarize the autoregressive generation costs for three model classes, Transformers, Convolutional Models, and RNNs, considering a prompt length of $T$ and the generation of $K$ tokens by each model, with $L$ the length of the convolutional filters (i.e., the maximum sequence length). We show the runtimes and memory requirements for each of the listed models in Table 1.

**Attention.** Processing a prompt requires $O(T^2)$ time. However, token-by-token generation can be accelerated to $O(T + K)$ for each generated token via key-value caching, with a total of $O(T^2 + K(T + K)) = O(T^2 + K^2)$ operations and requiring $O(T + K)$ space [37, 28].

**Convolutional Model.** A Convolutional Model with $k$ convolutional filters will require $O(kN)$ operations to autoregressively generate a new output given $N$ inputs, and thus a naive autoregressive convolutional implementation will require $O(k \cdot K \cdot (T + K))$ operations to generate $K$ tokens. A more refined "Epoched Future Fill" algorithm with prompt prefilling can reduce this to $O(k \cdot (T \log T + K^{3/2}\sqrt{\log K}))$ to generate $K$ tokens. The "Continuous Future Fill" algorithm has theoretical guarantees of $O(k \cdot (T \log T + K \log^2 K))$ operations, although it suffers from numerical instability and has not been implemented or used in practice [1]. For the guarantees of the STU architecture, we require $k = O(\log L)$ and in practice we choose $k = 24$ [22].

**RNN (LDS).** For an RNN with state dimension $h$, autoregressive generation requires $O(h)$ operations per token generated and $O(h)$ memory, allowing generation of $K$ tokens with $O(h \cdot (T + K))$ operations. As we will prove, for an LDS with representation capacity comparable to an STU with $k$ filters, we require $h = O(k) = O(\log L)$.

| Method | Prefill + Generation Runtime | Cache Size | Runtime with $K, T = O(L)$ |
|---|---|---|---|
| Standard Conv | $(TK + T \log T + K^2)k$ | $T + K$ | $L^2 \log L$ |
| Standard Attn. | $T^2 + K^2$ | $T + K$ | $L^2$ |
| EpochedFF | $(T \log T + K^{3/2}\sqrt{\log K})k$ | $K$ | $L^{3/2}(\log L)^{3/2}$ |
| ContinuousFF | $(T \log T + K \log^2 K)k$ | $K$ | $L \log^3 L$ |
| **SpectraLDS** (ours) | $(T + K)h$ | $h$ | $L \log L$ |

Table 1: Comparison of architecture runtime and memory requirements for generating $K$ tokens from a length $T$ prompt with $k, h = O(\log L)$, where $O(\cdot)$ is omitted for brevity.

## 4 Problem Background

In this section, we survey the fundamentals relevant to our approach. First, we discuss linear dynamical systems and the inherent challenges of training them directly on tasks requiring long memory. We then present the main theoretical results of spectral filtering and outline how the Spectral Transform Unit (STU) leverages fixed spectral filters to model linear recurrences without explicitly learning the transition matrix $A$. Finally, we set the stage for our method of distilling STUs back into linear dynamical systems.

### 4.1 Linear Dynamical Systems

Linear dynamical systems (LDS) have been widely used in control theory to represent time-dependent processes, forming the basis of classical state-space formulations and optimal control methods [20, 23, 3, 19, 6]. Concretely, we consider an input sequence $u_1, u_2, \ldots, u_t \in \mathbb{R}^n$, and the corresponding output sequence $y_1, y_2, \ldots, y_t \in \mathbb{R}^m$. The hidden state $x_t \in \mathbb{R}^d$ summarizes the system's memory of past inputs, with the evolution of the system being represented as

$$x_t = Ax_{t-1} + Bu_t, \quad y_t = Cx_t + Du_t. \tag{2}$$

where $A \in \mathbb{R}^{d \times d}$, $B \in \mathbb{R}^{d \times n}$, $C \in \mathbb{R}^{m \times d}$, and $D \in \mathbb{R}^{m \times n}$. We omit dynamics and observation noise terms for simplicity with this derivation, although we test our methods on signals with noise.

**Expanding the LDS to the Convolutional Form.** In a noiseless environment with $x_0 = \vec{0}$, we can expand the LDS equations as follows:

$$y_t = Cx_t + Du_t = C\Big(Ax_{t-1} + Bu_t\Big) + Du_t = \cdots = \sum_{i=0}^{t-1} CA^i B\, u_{t-i} + Du_t.$$

Note that if any eigenvalue $|\lambda_i(A)| > 1$, the system becomes unstable and $y_t$ may tend to infinity in magnitude. Even for $|\lambda_i(A)| < 1$, systems with $\|A\| \approx 1$ are prone to failure due to large $A^i$ powers in backpropagation, as a noisy algorithm may approximate $A$ with spectral radius greater than 1 during training. If $|\lambda_i(A)| < 1 - \delta$ for some spectral gap $\delta > 0$, then

$$y_t = \sum_{i=0}^{\tau} C A^i B\, u_{t-i} + \varepsilon_\tau, \quad \|\varepsilon_\tau\| \leq \varepsilon,$$

where $\tau = O\big(\frac{1}{\delta} \log \frac{1}{\varepsilon}\big)$ and the effective memory is thus on the order $\frac{1}{\delta}$ [2]. As $\delta \to 0$, learning $A$ directly becomes unstable for large contexts [5, 32, 30], highlighting the need for methods such as spectral filtering. Since the $D$ matrix serves as a skip connection that does not affect the representation capacity, we fix it to a 0-matrix and omit its consideration for the remainder of this paper. We will sometimes use the shorthand LDS$(C, A, B)$ to refer to a linear dynamical system with those parameters and $D = 0$.

Additionally, for the remainder of this paper, we restrict our attention to systems where $A$ is a symmetric real matrix. An LDS with symmetric $A$ can be diagonalized over the real numbers, making it equivalent to an LDS with a diagonal $A$. Without loss of generality, we therefore assume $A$ is diagonal with eigenvalues $\alpha_1, \ldots, \alpha_d$.

**Spectral Filtering.** With initial state $x_0 = \vec{0}$, defining $\mu(\alpha) = (1, \alpha, \alpha^2, \ldots)$, and indexing the columns of $C$ as $c_\ell$ and the rows of $B$ as $b_\ell$, we can extend the convolutional representation:

$$y_t = \sum_{i=0}^{t-1} C\, A^i\, B\, u_{t-i} = \sum_{i=0}^{t-1} C\left(\sum_{\ell=1}^{d} \alpha_\ell^i (e_\ell \otimes e_\ell)\right) B u_{t-i} = \sum_{\ell=1}^{d} (c_\ell \otimes b_\ell) \sum_{i=1}^{t} \mu(\alpha_\ell)(i) \cdot u_{t-i+1}$$

To circumvent the non-convex optimization problem of finding $\alpha$ that best fit an LDS, [14] propose the spectral filtering algorithm, which learns an approximation of $\mu(\alpha)$ in a convex manner. They prove that given eigenvalue-eigenvector pairs $\{\sigma_j, \phi_j\}_j$ of the Hankel matrix $Z$,

$$Z := \int_0^1 \mu(\alpha)\, \mu(\alpha)^\top \, d\alpha, \quad Z_{i,j} := \frac{2}{(i+j)^3 - (i+j)}$$

any $\mu(\alpha)$ with $0 \leq \alpha \leq 1$ can be approximated by the top $k$ eigenvectors $\{\phi_1, \ldots, \phi_k\}$ of $Z$ with an exponentially decreasing error in $k$. Thus, if $y_t$ is generated by a PSD linear dynamical system, we have the following result:

$$y_t \approx \sum_{\ell=1}^{d} (c_\ell \otimes b_\ell) \sum_{i=1}^{t} \tilde{\mu}(\alpha_\ell)(i) \cdot u_{t-i+1} = \sum_{j=1}^{k} M_j \left(\sum_{i=1}^{t} \phi_j(i) \cdot u_{t-i+1}\right)$$

where we define $\tilde{\mu}(\alpha) := \sum_{i=1}^{k} \langle \mu(\alpha), \phi_i \rangle \phi_i$ and learn suitable parameters $M_j \in \mathbb{R}^{m \times n}$. Rather than depending on powers of $A$, learning an LDS with this parameterization remains convex in $\{M_j\}$, since eigenvectors $\{\phi_j\}$ from the matrix $Z$ are computed offline.

**The Spectral Transform Unit.** To account for learning negative eigenvalues of $A$, the spectral filtering construction can be adapted by introducing positive and negative sets of feature maps. If $\{\sigma_j, \phi_j\}$ are the eigenvalue-eigenvector pairs of $Z$, then for each time $t$ and each respective filter $\phi_j$, we define the projections of the inputs onto the spectral basis:

$$U_{t,j}^+ = \sum_{i=1}^{t} u_{t-i+1} \cdot \phi_j(i), \qquad U_{t,j}^- = \sum_{i=1}^{t} u_{t-i+1} \cdot (-1)^{i-1} \cdot \phi_j(i).$$

One then forms the output by learning linear combinations of both $U_{t,k}^+$ and $U_{t,k}^-$ and an optional autoregressive term [12, 2]:

$$y_t^{\text{SF}} = \underbrace{\sum_{j=1}^{k} M_j^{\phi+} U_{t-2,j}^+ + \sum_{j=1}^{k} M_j^{\phi-} U_{t-2,j}^-}_{\text{Spectral Filtering component}} + \underbrace{\hat{y}_{t-2} + \sum_{i=1}^{3} M_i^u u_{t+1-i}}_{\text{AR component}} \tag{3}$$

Without the autoregressive component we compute $y_t^{\text{SF}} = \sum_{j=1}^{k} M_j^{\phi+} U_{t,j}^+ + \sum_{j=1}^{k} M_j^{\phi-} U_{t,j}^-$. The above expression is considered the *Spectral Transform Unit*, where $\{M_j\}$ is the set of parameters to be learned using a differentiable algorithm. Following [22], we consider the STU without the autoregressive component. Empirical evidence [22] shows that hybrid STU models can compete with or even outperform purely attention-based architectures.

**Error Bounds for Spectral Approximation.** We repeat the result from [14] (see also [2, 26, 27]) stating that given any LDS parameterized by $A, B, C$ where $A$ is a symmetric matrix with $\|A\| \leq 1$, there exist matrices $M_1^{\phi+}, \ldots, M_k^{\phi+}, M_1^{\phi-}, \ldots, M_k^{\phi-}$ such that for all $L$ and for all input sequences $u_1, \ldots, u_L$, with $\|u_t\| \leq 1$, the following holds for all $t \in [L]$:

$$\|y_t^{\text{LDS}} - y_t^{\text{SF}}\| \sim e^{-\frac{k}{\log L}}.$$

where $k$ is the number of spectral filters, $y_t^{\mathrm{LDS}}$ is the sequence generated by the LDS, and $y_t^{\mathrm{SF}}$ is the sequence generated by spectral filtering.

Therefore, one can approximate any LDS that meets these specifications up to error $\varepsilon$ by selecting $k = O\big(\log L \log\big(\frac{1}{\varepsilon}\big)\big)$ spectral filters. Thus, the STU can capture LDS dynamics with only a logarithmic number of filters $k$, providing a compact and stable representation even for systems with high effective memory ($\|A\| \approx 1$).

## 4.2   Distilling STU into an LDS

While the STU avoids direct learning of $A$, it still implements LDS-like dynamics via spectral filtering. A natural question is whether we can recover explicit parameters $(\widetilde{A}, \widetilde{B}, \widetilde{C})$ from the learned STU, enabling an equivalent recurrence rather than $O(L)$ costs when convolving spectral filters with input sequences. This can be achieved by approximating the convolution kernel of the STU by the implicit convolution kernel of an LDS. Such a distillation would bridge the gap between the stable convex training of the STU and the fast inference of a recurrent LDS.

# 5   Algorithm and Main Result

We now present our main theoretical result and accompanying algorithm, which shows how to recover an accurate LDS representation from the learned spectral filters. Concretely, we demonstrate that each STU filter $\phi_j$ can be approximated by a linear combination of geometrically decaying LDS impulse response filters (i.e., the LDS implicit convolutional kernel).

## 5.1   A General Transformation from Spectral Filters to LDS

Our main result is given in Algorithm 2. For the dynamics $\mathrm{LDS}(1 - \alpha, \alpha, 1)$, we denote the impulse response filter by
$$\mu_L(\alpha) = (1 - \alpha)[1 \quad \alpha \quad \alpha^2 \quad \ldots \quad \alpha^{L-1}].$$
Note that $\mu_L(\alpha) * u_{1:L} = \sum_{i=1}^{L}(1 - \alpha)\alpha^{i-1} \cdot u_{L-1} = \mathrm{LDS}(1 - \alpha, \alpha, 1)(u_{1:L})$, and thus inference with the LDS is the same as convolution with its impulse response filter. Let $\phi_1, \ldots, \phi_k \in \mathbb{R}^L$ be the spectral filters of length $L$. We write the first $k$ filters in matrix form as $\Phi_{1:k} \in \mathbb{R}^{k \times L}$, such that the $i$-th row is $\phi_i$.

A first observation is that we can write any LDS impulse filter approximately in the spectral basis. This is a direct consequence of the spectral filtering methodology for learning an LDS [14].

---

**Algorithm 1** FindSpectralRepresentation

1: **Input:** Scalar LDS parameter $\alpha \in \mathbb{R}$, representation size $k$.
2: **Output:** Spectral parameters $m \in \mathbb{R}^k$.
3: Construct the impulse response vector $\mu_L(\alpha) \in \mathbb{R}^L$ as $\mu_L(\alpha) = (1 - \alpha)[1, \alpha_i, \alpha_i^2, \ldots, \alpha_i^{L-1}]$.
4: **return** best spectral fit of the system over random inputs $u \in \mathbb{R}^L$ using gradient descent

$$m = \arg \min_{m \in \mathbb{R}^k} \mathbb{E}_{u \in \mathbb{R}^L} \left[ \left| m^\top \Phi_{1:k} u - \mu_L(\alpha)^\top u \right|^2 \right].$$

---

As a consequence of results from spectral filtering for learning LDSs, we show in Appendix A.4 that the procedure FindSpectralRepresentation returns a vector $m$ for which, for some constant $c > 0$,
$$\left\| m^\top \Phi_{1:k} - \mu_L(\alpha) \right\| \leq c \, e^{-\frac{k}{\log L}}.$$

We proceed to use this subroutine to find a distillation to the spectral filters.

Our main performance guarantee is given in the following theorem.

**Theorem 1.** *As long as $h \geq k$, Algorithm 2 returns w.h.p. a matrix $\widetilde{M}$ such that*
$$\left\| \Phi_{1:k} - \widetilde{M} \, \mu_L(\alpha_1, \ldots, \alpha_h) \right\| \leq c \, \lambda_{\max} h \, e^{-\frac{k}{\log L}},$$
*where $\lambda_{\max}$ is the largest eigenvalue of the Penrose-Moore pseudo inverse of the matrix $M$.*

---
**Algorithm 2** Spectral Filters to LDS Filters
---
1: **Input:** The first $k$ spectral filters matrix $\Phi_{1:k} \in \mathbb{R}^{k \times L}$; and parameter $h > k$.
2: **Output:** The transformation matrix $\widetilde{M}$.
3: Sample $h$ randomly chosen independent scalars $\alpha_1, ..., \alpha_h$ with each $\alpha \in [0, 1]$.
4: Construct the matrix $\mu_L(\alpha_{1:h}) \in \mathbb{R}^{h \times L}$ with $i$th row $\mu_L(\alpha_i) = (1 - \alpha_i)[1, \alpha_i, \alpha_i^2, \ldots, \alpha_i^{L-1}]$.
5: For each scalar impulse response, find the spectral representation by

$$m_i = \text{FindSpectralRepresentation}(\alpha_i).$$

6: Let $M$ be the $h \times k$ matrix whose rows are $m_i$.
7: **return** $\widetilde{M} := M^{-1} \in \mathbb{R}^{k \times h}$
---

The significance of Theorem 1 is that it allows us to translate between the representation of spectral filters and linear dynamical systems.

We note that it is not immediate to upper bound $\lambda_{\max}$. Indeed, for $h \sim k$, this can be exponentially large in $k$, as it corresponds to the condition number of a Vandermonde matrix [4]. However, we note experimentally, under the distribution described in the practical algorithm below, that as $h$ grows, $\lambda_{\max}$ quickly becomes smaller. This is an overparametrization effect, which we show experimentally in Appendix A.3. We provide analysis of Theorem 1 in Appendix A.4.

### 5.2 Practical Algorithm

For improved practical performance we adopt the following procedure. Fix $H \gg h$. As above, let $\mu_L(\alpha_{1:H}) \in \mathbb{R}^{H \times L}$ denote $H$ impulse responses and let $\Phi_{1:k} \in \mathbb{R}^{k \times L}$ denote the first $k$ spectral filters. To obtain a representation of size $h$, we select a small index set $S \subseteq \{1, \ldots, H\}, |S| = h \ll H$, by running multi-target Orthogonal Matching Pursuit (OMP) [25, 36] on the regression

$$\mu_L(\alpha_{1:H})^\top X \approx \left(\Phi_{1:k}\right)^\top.$$

OMP greedily sets one coefficient per step from 0 to reduce the regression loss, with the selected coefficients after $h$ steps providing the indices for $S$. Given $S$, we solve the unregularized least-squares problem

$$X^\star = \arg\min_X \left\| \mu_L(\alpha_S)^\top X - \left(\Phi_{1:k}\right)^\top \right\|_F^2,$$

to yield $\Phi_{1:k} \approx \widetilde{M} \mu_L(\alpha_{1:H})$, where $\widetilde{M} = (X^*)^\top$.

Another change is the distribution from which the $\alpha_i$ are drawn. We employ symmetric, near-unit-radius sampling to emphasize long-memory behavior: for each $i$, draw $U_i \sim \text{Unif}[0, 1]$ and $S_i \sim \text{Unif}(\{-1, 1\})$, and set

$$\alpha_i = S_i\left(1 - U_i^4\right).$$

Equivalently, $|\alpha_i| \stackrel{d}{=} 1 - U_i^4$ with an independent random sign. This deliberately skews mass toward $|\alpha_i| \approx 1$, improving reconstruction at a better $h$ to $H$ ratio. We ablate this choice in Appendix A.9. For numerical stability we use `float64` arithmetic end-to-end and scale $\Phi_{1:k}$ by $s_{\text{num}} > 0$ so that $\|\mu_L(\alpha_{1:H})\|_F \approx \|s_{\text{num}}\Phi_{1:k}\|_F$. To mitigate the impact of magnitude on OMP, we normalize the columns of $\mu_L(\alpha_{1:H})^\top$. We later undo the scaling. In Figure 1 we report the reconstruction error of $\Phi_{1:k}$ against an LDS of hidden dimension $h$ produced by this practical method.

### 5.3 Converting $\widetilde{M}$ into an LDS.

As a result of Theorem 1, with $A := \text{Diag}(\alpha_1, \ldots, \alpha_h)$, $\Gamma := \text{Diag}(1 - \alpha_1, \ldots, 1 - \alpha_h)$ and $\mu := \mu_L(\alpha_1, \ldots, \alpha_h)$, we can replace the costly STU convolutions:

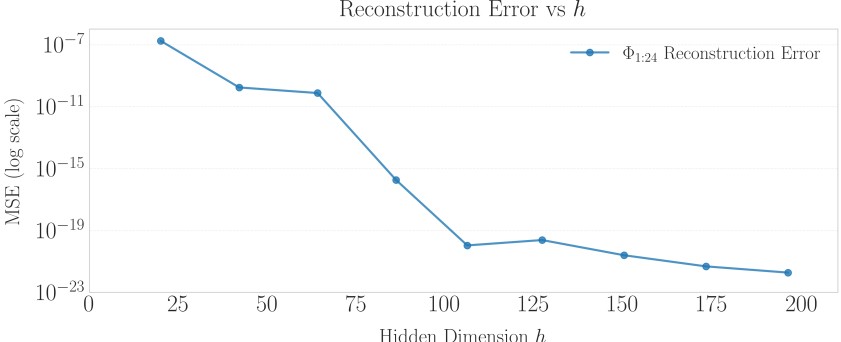

Figure 1: Reconstruction error of spectral filters as a function of LDS state dimension. For this experiment, $H = 10000$, but results are similar for $H = 1000$ to $20000$ (see Appendix A.8).

$$U_{t,j}^+ = \sum_{i=1}^{t} u_{t-i+1} \cdot \phi_j(i) \approx \sum_{i=1}^{t} \left( \widetilde{M}_j \mu \right)(i) \cdot u_{t-i+1} = \widetilde{M}_j \sum_{i=0}^{t-1} \Gamma A^i \, \mathbf{1}_h \, u_{t-i}^\top = \text{LDS}(\widetilde{M}_j \Gamma, A, \mathbf{1}_h)(u_{1:L}^\top)$$

$$U_{t,j}^- = \sum_{i=1}^{t} u_{t-i+1} \cdot \phi_j(i) \approx \widetilde{M}_j \sum_{i=0}^{t-1} \Gamma(-A)^i \, \mathbf{1}_h \, u_{t-i}^\top = \text{LDS}(\widetilde{M}_j \Gamma, -A, \mathbf{1}_h)(u_{1:L}^\top)$$

This provides the basis of our autoregressive inference advantage: rather than computing $U_{t,j}^+$ and $U_{t,j}^-$ as pure convolutions, we can maintain the hidden state for $\text{LDS}(\widetilde{M}_j \Gamma, A, \vec{1})$ and $\text{LDS}(\widetilde{M}_j \Gamma, -A, \vec{1})$ for the $O(h)$ computation during inference. For practical efficiency, we can compute all $U_{t,1}^+, \ldots, U_{t,k}^+, U_{t,1}^- \ldots U_{t,k}^-$ simultaneously with a single LDS by leveraging the similarities in the state updates (see Appendix A.7).

### 5.4 LDS to LDS Distillation

A direct consequence of our approach is that we can distill any high-dimensional symmetric LDS into a low-dimensional LDS with a bounded error. For an LDS with input dimension $d_{in}$ and output dimension $d_{out}$, spectral filtering provides an $\varepsilon$-approximation with only $O(d_{in} \cdot d_{out} \cdot \log L \cdot \log(\frac{1}{\varepsilon}))$ parameters regardless of the hidden dimension.

Accepting that $h\lambda_{max}$ is $O(1)$ for $h \gg k$, as justified in Appendix A.3, we can then convert this spectral representation by application of our distillation procedure into an LDS with state dimension $d_{in} \cdot h$, where with $h = O(k) = O(\log L \log\left(\frac{1}{\varepsilon}\right))$ we maintain $\varepsilon$ error. Thus, the distilled LDS has $O(d_{in} \cdot d_{out} \cdot \log L \cdot \log(\frac{1}{\varepsilon}))$ total parameters. In other words, the combination of spectral filtering for LDS learning and the following distillation step yields a practical method to reduce the state dimension while preserving the system's dynamics within tight error bounds. Empirically, for tasks as difficult as modeling language, we only require $k \leq 24$ filters for strong performance, and the state dimension $h \geq 80$ of the distilled LDS suffices to retain performance.

## 6 Experiments

To demonstrate the effectiveness of our distillation algorithm, we begin by illustrating that a low-dimensional LDS can effectively approximate the spectral filters in Figure 3 and further examine the eigenvalues of the resulting system in Appendix A.7. Building on this, to further quantify the effectiveness of Algorithm 2 in fitting the spectral filters with practically efficient LDSs, we plot the reconstruction error of the spectral filters across different LDS state dimensions in Figure 1. These results show that our algorithm, guided by Theorem 1 amply reduces the approximation error to a sufficiently small level without an excessive increase in state dimension.

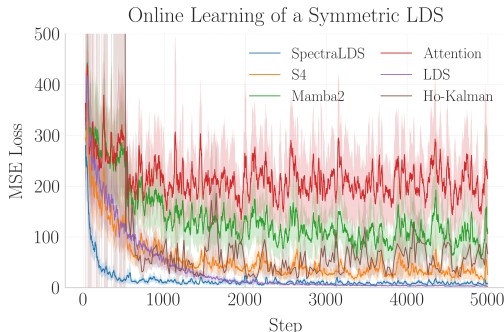
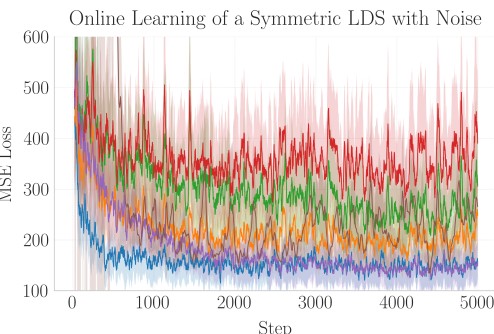

Figure 2: Comparison of SpectraLDS and other methods learning an arbitrary symmetric LDS with and without noise. The shaded region shows the $95\%$ confidence interval over $8$ runs. Each model leveraged default configurations except the LDS, which required a lower learning rate to converge. More details are available in Appendix A.5.

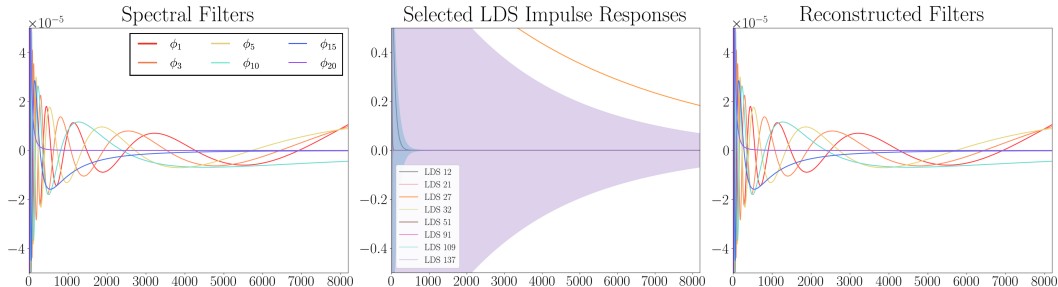

Figure 3: Fit of Spectral Filters by an LDS of state dimension 160, where x-axis represents the time domain. 80 dimensions are for the spectral filters ($U_{t,k}^{+}$) and 80 for the negative features ($U_{t,k}^{-}$, not shown). Filters are normalized to be comparable. The shading on the middle figure represents a filter quickly alternating (negative $\alpha$). A complete comparison for $k = 24$ without normalization is provided in Appendix A.6. These filters have MSE error $7.689 \times 10^{-19}$.

To validate our method quantitatively, we perform synthetic experiments comparing SpectraLDS against other benchmarks on learning high-memory LDSs (see Appendix A.10). We find that our method significantly outperforms strong baselines in both sample efficiency and reconstruction accuracy, confirming that our approach is well suited for learning systems with long-range dependencies.

Turning to the large-scale evaluation, we distill a 340M-parameter FlashSTU model [22] into an LDS-based architecture and compare its performance across a suite of language benchmarks. From the results in Table 2, we point out that despite the change from convolution-based spectral filters to an explicit LDS representation for the STU layers, the performance remains identical across all tasks. This observation supports our claim that the STU can be closely approximated by a low-dimensional LDS without compromising predictive accuracy. We provide details of the experimental setup and hyperparameters for the models used in Appendix A.13.

| Model | MMLU | Hella. | PIQA | BoolQ | Wino. | CSQA | OBQA | ARC-e | ARC-c | Average |
|---|---|---|---|---|---|---|---|---|---|---|
| Flash STU 340M | 26.58 | 30.46 | 65.34 | 60.12 | 51.85 | 20.48 | 20.60 | 54.08 | 23.29 | 39.20 |
| SpectraLDS 340M | 26.58 | 30.45 | 65.29 | 60.12 | 50.99 | 20.15 | 20.20 | 54.17 | 23.29 | 39.03 |
| Flash STU Std. Err. | 0.37 | 0.47 | 1.11 | 0.86 | 1.40 | 1.16 | 2.06 | 1.02 | 1.24 | – |
| SpectraLDS Std. Err. | 0.37 | 0.46 | 1.11 | 0.86 | 1.40 | 1.15 | 2.07 | 1.02 | 1.24 | – |
| Transformer 340M | 26.81 | 30.41 | 64.64 | 61.10 | 51.62 | 19.98 | 18.80 | 55.47 | 21.84 | 38.96 |

Table 2: Evaluation of a 340M-parameter FlashSTU model and its distilled representation, replacing each convolution with an LDS of state dim. 160, on language benchmarks. Despite converting convolutions into an explicit LDS formulation, performance remains statistically equivalent.

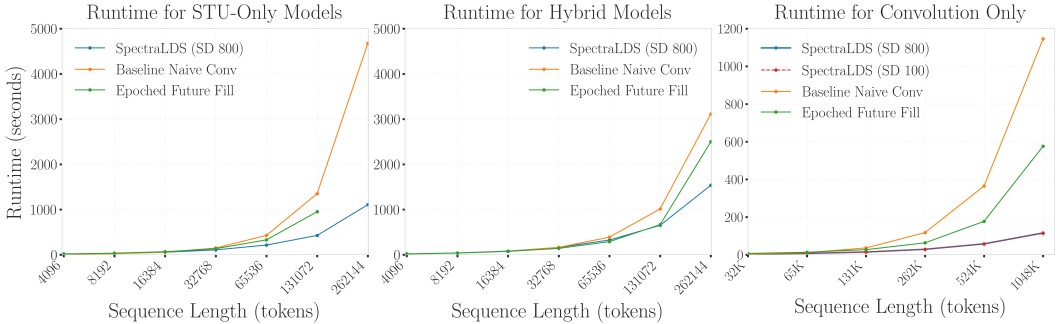

Figure 4: Runtime for generating sequences of increasing length across STU implementations. The naive convolution approach exhibits quadratic growth, the FutureFill variants show logarithmic growth, and the distilled STU-to-LDS layers achieve linear growth. The STU-Only Epoched Future Fill OOMs for the largest sequence length. As shown in the rightmost figure, the SpectraLDS models have nearly identical runtime despite varied state dimension. Further results are in Appendix A.11.

Finally, we present various measures of inference speed to illustrate the constant-time per token generation provided by the distilled STU. In Figure 4, we compare the inference speed of a distilled STU model against a naive convolutional approach and a numerically stable FutureFill variant [1]. In a 12 STU layer model, the naive convolution exhibits quadratic runtime growth with sequence length; the FutureFill variants achieve logarithmic growth; and the distilled STU-to-LDS model demonstrates the best performance with linear growth.

In the hybrid model with 6 attention and 6 STU layers, we find the distilled LDS implementation still provides a significant performance increase. Additionally, we note increases in the LDS state dimension have little impact on the overall runtime, indicating that LDS operations are not a compute bottleneck (see Appendix A.11). For both the hybrid and STU-only models up to $16,384$ tokens, the distilled LDS, naive convolution, and Epoched Future Fill all have similar runtimes. All performance benchmarks were conducted on a single H100 GPU, with each generation process evaluated separately to ensure consistent measurements. We provide full experiment details in Appendix A.12.

Looking ahead, further investigation is warranted to better understand how convolution, LDS, and attention layers interact at the hardware level, and to optimize their coordination for even greater speedup. Additionally, further work is required to determine if the LDS layer is stable below float64.

To conclude, since optimized transformer implementations suffer from KV-Cache memory bottlenecks rather than compute bottlenecks [9], and the LDS layers have drastically lower memory requirements, we anticipate that, **with appropriate optimization, the inference speed of hybrid attention-STU architectures will be independent of the amount of LDS layers.**

# 7   Conclusion and Discussion

We have provided the first provable technique for learning the parameters of a symmetric LDS with arbitrarily high effective memory. By leveraging their convex-learning approach, we show how spectral filters can be distilled into an explicit LDS representation, enabling the construction of a state-space model with logarithmic-time inference and theoretical guarantees on the loss bounds.

We have provided a lower bound on how increasing state dimension affects reconstruction loss for a linear dynamical system, and we envision SpectraLDS as a drop-in replacement for autoregressive convolutional layers in certain sequence-to-sequence tasks.

# 8   Acknowledgments

EH gratefully acknowledges support from the Office of Naval Research and Open Philanthropy.

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

# A    Appendix

## A.1    Notation throughout the paper

| | |
|---|---|
| $x_t \in \mathbb{R}^d$ | state at time $t$ |
| $u_t \in \mathbb{R}^n$ | control (input) at time $t$ |
| $y_t \in \mathbb{R}^m$ | sequence model outputs at time $t$ |
| $A, B, C, D$ | system matrices for linear dynamical system |
| $h$ | Number of $\alpha$ used in final LDS representation |
| $H$ | Number of random samples of $\alpha$ |
| $k$ | Number of spectral filters |
| $K$ | Length of generated sequence |
| $T$ | Length of prefill sequence |
| $L$ | Length of sequence (or of filters) |

## A.2    Experimental details and notation

All experiments were performed on Nvidia H100-80GB GPUs in PyTorch [33]. All layers except the LDS leverage bfloat16 precision, whereas the LDS layers require float64 precision. All tests, unless otherwise stated, use $k = 24$ spectral filters and replace the $k$ filters with an LDS with state-dimension $h = 80$. To fit the additional 24 alternating filters of the negative component (i.e., $\phi_j^-$, where $\phi_j^-[i] = (-1)^{i-1}\phi_j[i]$, as is needed to compute $U_{t,j}^-$ purely convolutionally), we expand to an LDS with state-dimension $h = 160$.

For the layer-level inference speed benchmarks, we leverage the STU with tensor-dot approximation (STU-T) [2], the same variant used in the FlashSTU language model [22]. It is worth noting that the STU-to-LDS distillation leads to LDS layers with comparable speed regardless of whether the tensor-dot approximation is employed, and thus this approximation only accelerates the STU layers in benchmarks. A formal definition of this approximation is as follows:

**STU Tensor-Dot Approximation:**    [2] introduced an important optimization, the tensor-dot approximation, wherein each tensor $M \in \mathbb{R}^{d_{\text{in}} \times 2k \times d_{\text{out}}}$, representing a concatenation of the tensors $M_1^{\phi+}, \ldots, M_k^{\phi+}, M_1^{\phi-}, \ldots, M_k^{\phi-}$, is learned as $M^{(1)} \times M^{(2)}$ for $M^{(1)} \in \mathbb{R}^{d_{\text{in}} \times 2k}$ and $M^{(2)} \in \mathbb{R}^{d_{\text{in}} \times d_{\text{out}}}$.    This approximation allows for a reduction in convolutions as, with input $x_1, x_2, \ldots, x_\ell$, we can compute $y_\ell^{\text{SF}} \approx \sum_{i=1}^{\ell} (x_{\ell-i+1}^\top M^{(2)}) \odot M_{\text{filters}}[i]$, where $M_{\text{filters}} = [\phi_1, \ldots, \phi_k, \phi_1^-, \ldots, \phi_k^-]^\top M^{(1)} \in \mathbb{R}^{L \times d_{\text{out}}}$ and $\odot$ refers to the Schur product (i.e. $(x \odot y)_j = x_j \cdot y_j$). This allows for only $d_{\text{out}}$ convolutions with the tensor-dot approximation, as opposed to $k \cdot d_{\text{in}}$ convolutions without it. Although this method can reduce expressivity and does not inherit the same marginal-stability guarantees, it maintains competitive empirical performance while yielding significant improvements in efficiency.

Additionally, we frequently refer to the impulse response of the LDS and STU models. The impulse response of a linear sequence model $f : \mathbb{R}^L \to \mathbb{R}^1$ is the vector or convolutional kernel $\psi \in \mathbb{R}^L$ that is equivalent to $f$ (i.e. $f(x_{[1,\ldots L]}) = \psi * x_{[1,\ldots L]}$), and thus:

$$\psi[t] = f(\underbrace{[0, \ldots, 1, \ldots, 0]}_{\text{1 at position } L-t+1})$$

We only consider the impulse response of $f : \mathbb{R}^L \to \mathbb{R}^1$, but the impulse response is closely related to the derivative with respect to the inputs and is generalized identically. For an STU model with $d_{\text{in}} = d_{\text{out}} = 1$ and parameters $M_1^{\phi+}, \ldots, M_k^{\phi+}, M_1^{\phi-}, \ldots, M_k^{\phi-} \in \mathbb{R}$, the impulse response $\psi_{\text{SF}}$ is thus described by $\psi_{\text{SF}}[t] = \sum_{j=1}^k M_j^{\phi+} \phi_j[t] + (-1)^{t-1} \sum_{j=1}^k M_j^{\phi-} \phi_j[t]$. Similarly, for an LDS with $d_{\text{in}} = d_{\text{out}} = 1$ and parameters $a, b, c \in \mathbb{R}$, the impulse response $\psi_{\text{LDS}}$ is described by $\psi_{\text{LDS}}[t] = ca^{t-1}b$

## A.3 Experimental Results on the Condition Number of $M$, the Spectral Coefficients Matrix

In Figure 5, we present experimental results on the condition number of $M$ as defined in Section 5. Recall that $M \in \mathbb{R}^{h \times k}$ is the spectral coefficients matrix produced by Algorithm 2; its $i$-th row $m_i^\top$ stores the coefficients that express the LDS impulse response $\mu_L(\alpha_i)$ (with geometric decay factor $\alpha_i$) in the spectral basis $\Phi_{1:k}$. Starting at $h = k$ with $k = 24$, we repeatedly add additional independent vectors $\alpha_i$ and measure the maximum singular value of the constructed $M^{-1}$. The blue central line shows the mean maximum singular value across 5 experiments for a given value of $h$, with the shaded region showing the maximum and minimum of the largest singular values across experiments. We draw $\alpha$ independently from the distribution in Figure 7. To justify our statement in section 5.2 that $\lambda_{max} \cdot h$ can be considered $O(1)$, we additionally plot $\lambda_{max} \cdot h$ under the same experimental setup in Figure 6.

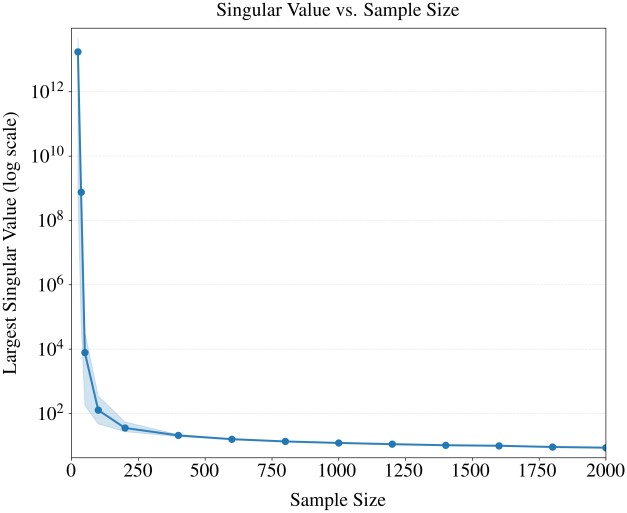

Figure 5: Largest Singular Value as we increase $h$.

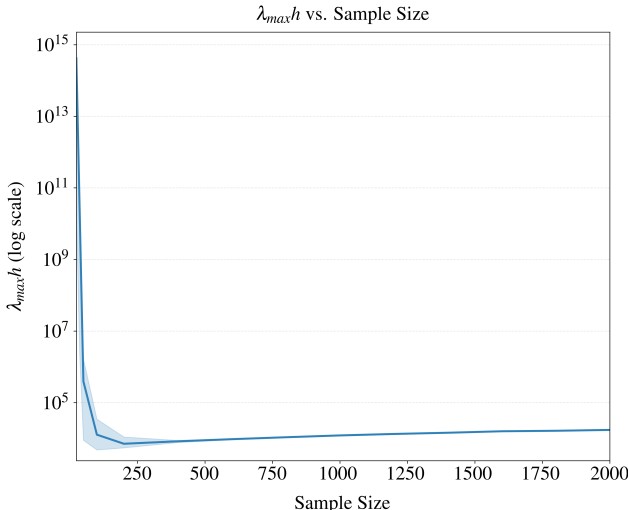

Figure 6: $\lambda_{max} \cdot h$ as we increase $h$.

## A.4 Analysis of Theorem 1

*Proof of Theorem 1.* Let

$$\mu_L(\alpha) = (1 - \alpha) \begin{bmatrix} 1 & \alpha & \alpha^2 & \ldots & \alpha^{L-1} \end{bmatrix} \in \mathbb{R}^L,$$

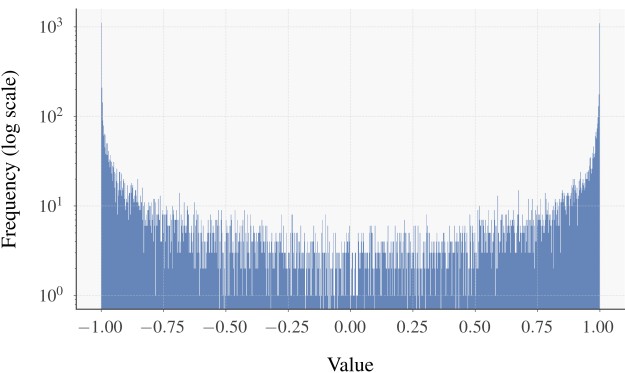

Figure 7: Distribution of $\alpha$ ($H = 10{,}000$ samples)

be the $L$-impulse response of the one dimensional linear dynamical system with parameters $\alpha, b = 1, c = 1 - \alpha$. Lemma 11.3 from [13] asserts that for any $\alpha \in [0, 1]$, there exist real coefficients $m_1, \ldots, m_k \in \mathbb{R}^k$ and constant $c > 0$, such that any sequence of inputs $u_{L:1} \in \mathbb{R}^L$ with $\|u_{L:1}\|_\infty \leq 1$,

$$\left| \sum_{i=1}^{k} m_i \langle \phi_i, u_{L:1} \rangle - \langle \mu_L(\alpha), u_{L:1} \rangle \right| \leq c e^{-\frac{k}{\log L}}.$$

The RHS term represents the evolution of the LDS, with system matrices $b, c$ that are assumed to be identity, and $\mu_L(\alpha)$ is the system evolution.

As the above is true for all $u_{L:1}$, for each $\alpha_j$, there exist $m_j$ such that,

$$\forall j \quad , \quad \left\| m_j^\top \Phi_{1:k} - \mu_L(\alpha_j) \right\| \leq c e^{-\frac{k}{\log L}}. \tag{4}$$

Moreover, as the optimization problem is convex in $m_j$ and Lipschitz continuous, FindSpectralRepresentation will return such $m_j$.

Let $M \in \mathbb{R}^{h \times k}$ be the matrix whose rows are $m^j \in \mathbb{R}^k$:

$$M = \begin{pmatrix} - m^1 - \\ - m^2 - \\ \vdots \\ - m^h - \end{pmatrix}.$$

Let $\mathcal{E} = M \Phi_{1:k} - \mu_L(\alpha_1, \ldots, \alpha_h)$. By the triangle inequality and (4),

$$\|\mathcal{E}\|_1 \leq \sum_{j=1}^{h} \left\| m_j^\top \Phi_{1:k} - \mu_L(\alpha_j) \right\| \leq c h e^{-\frac{k}{\log L}}. \tag{5}$$

Thus, assuming that $M$ is full rank, multiplying $\mathcal{E}$ by the Penrose-Moore pseudo-inverse of $M$ and using Holder's inequality, we get

$$\left\| \Phi_{1:k} - M^{-1} \mu_L(\alpha_1, \ldots, \alpha_h) \right\| = \|M^{-1} \mathcal{E}\| \leq \|M^{-1}\|_\star \|\mathcal{E}\|_1 \leq \lambda_{\max} \cdot c h e^{-\frac{k}{\log L}}.$$

It remains to argue that $M$ is full rank. This follows since $\Phi_{1:k}$ is an orthogonal basis, and the matrix $\mu_L(\alpha_{1:h})$ is a Vandermonde matrix. Thus, both matrices are full rank.

$\square$

## A.5 Learning a Symmetric LDS with and without noise

Figure 2 in the main paper compares SpectraLDS to other system-identification methods on the task of learning an arbitrary symmetric LDS with and without noise. The LDS signal has hidden

dimension 256, input and output dimension 5, and maximum eigenvalue magnitude 0.99 with Gaussian initialization. Each step provides a Gaussian input sequence of length 100 with variance $1/d_{in}$ and the final output. For the learning with noise experiment, Gaussian noise with 0.5 variance was added to the hidden state at each step, and Gaussian noise with 5 variance was added to the output. Each architecture was trained with the default PyTorch RMSProp optimizer configuration, except for the LDS, which required a lower learning rate to converge. The Ho-Kalman matrices are recomputed every 20 steps for computational ease, while the other methods were updated every step, and the Ho-Kalman parameters are the maximum for an input of length 100 (that is T_horizon = 99, T1 = 49, T2 = 49, state_dim_est = 48). The shaded region in the figure shows the 95% confidence interval over 8 runs.

## A.6    Fit of the Spectral Filters with a Linear Dynamical System (State Dimension 160)

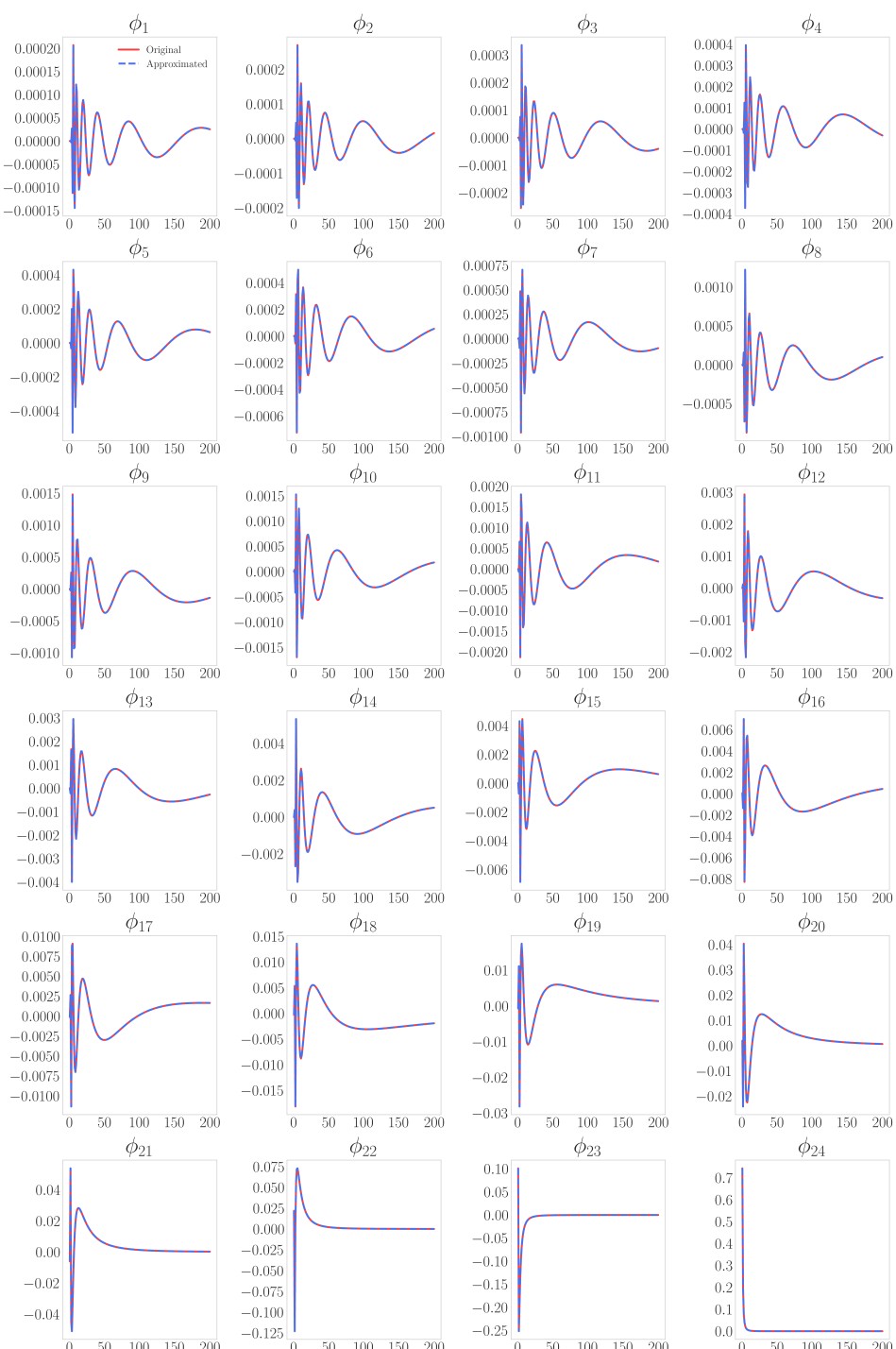

Figure 8: Visualization of the positive spectral filters and their approximation using an LDS with state dimension 160, as obtained via the practical algorithm. As described previously, only the first 80 dimensions of the state are used for approximate the Spectral Filters, with the remaining used for the alternating filters (negative filter maps), as described in Appendix A.7. The figure illustrates that even with a low-dimensional state, the LDS can accurately fit the spectral filters, confirming the efficacy of our distillation process.

## A.7 State Dynamics of the Linear Dynamical System

We train a linear dynamical system with state dimension 160 to fit the Spectral Filters and alternating filters using Algorithm 2 and examine the eigenvalues of the resulting system in Figure 9. We only plot the 80 eigenvalues corresponding to the Spectral Filters, as the remaining 80 corresponding to the alternating filters are identical except multiplied by $-1$ (see below the plot). The reconstruction error of the spectral filters for the trained LDS is $7.69 \times 10^{-19}$. We transform the eigenvalues to visualize the full range of the values as they approach 1 and differentiate the negative and positive eigenvalues as blue and red (e.g, $0.97$ and $-0.97$ would both map to $\approx 3.5$, although the first would join a red column and the second blue).

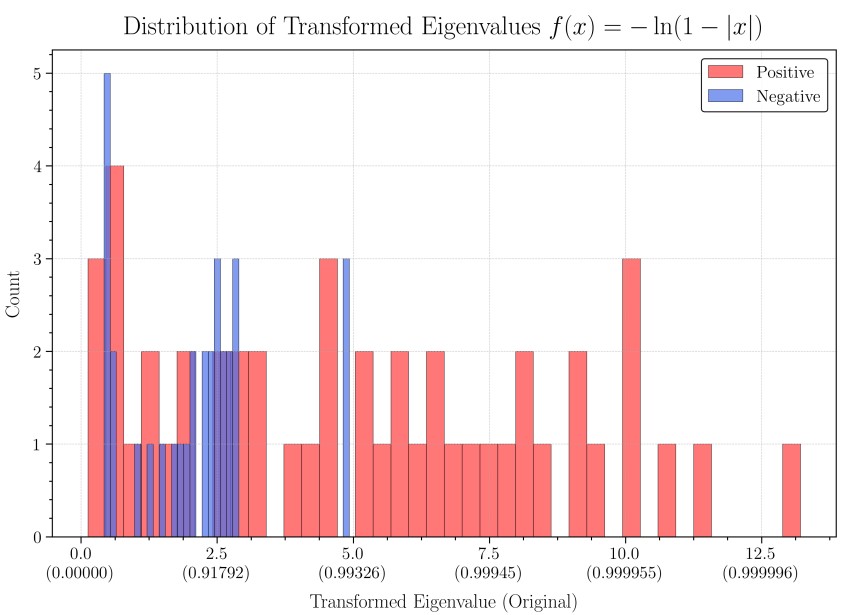

Figure 9: Distributions of Eigenvalues for LDS corresponding to the first 24 Spectral Filters.

For practical efficiency, we can compute all $U_{t,1}^+, \ldots, U_{t,k}^+, U_{t,1}^- \ldots U_{t,k}^-$ simultaneously with a single LDS parameterized by $\left\{ \begin{bmatrix} \widetilde{M}\,\Gamma & \mathbf{0} \\ \mathbf{0} & \widetilde{M}\,\Gamma \end{bmatrix}, \begin{bmatrix} A & \mathbf{0} \\ \mathbf{0} & -A \end{bmatrix}, \mathbf{1}_{2h} \right\}$. This LDS has a state dimension of $2hm$, where each of the $m$ input dimensions have a hidden state updated by $\begin{bmatrix} A & \mathbf{0} \\ \mathbf{0} & -A \end{bmatrix}$. The hidden state dimension differs from $2h$ as we treat $m$ as a batch dimension, resulting in $Bu_t \in \mathbb{R}^{2h \times m}$ rather than $\mathbb{R}^{2h}$. This mimics an LDS acting independently across each of the $m$ dimensions, equivalent to treating $m$ as a batch-axis, and this is the analog to how each of the $k$ spectral filters convolve along each input dimension independently. The above Figure 9 showcases the eigenvalues of matrix $A$, whereas for practical deployment, the state matrix is $\begin{bmatrix} A & \mathbf{0} \\ \mathbf{0} & -A \end{bmatrix}$.

## A.8 Reconstruction Error of the Spectral Filters varying LDS State Dimension and $H$

We demonstrate the efficacy of the practical algorithm in fitting the Spectral Filters with a low initial LDS State Dimension. In Figure 10, we ablate the choices of $H$ and $h$ for the practical algorithm. We do not fully understand why larger $H$ leads to worse fits for small state dimension. Notably, the error quickly becomes no longer human perceptible. In this procedure we only aim to fit the $k$ positive spectral filters. To fit both the $k$ positive spectral filters and $k$ negative spectral filters, we can achieve the same error with double the state dimension, as described in Appendix A.7. With our implementation of sparsity, the resulting hidden dimension will be near but not exactly that requested, which is why each line has different reconstruction dimensions.

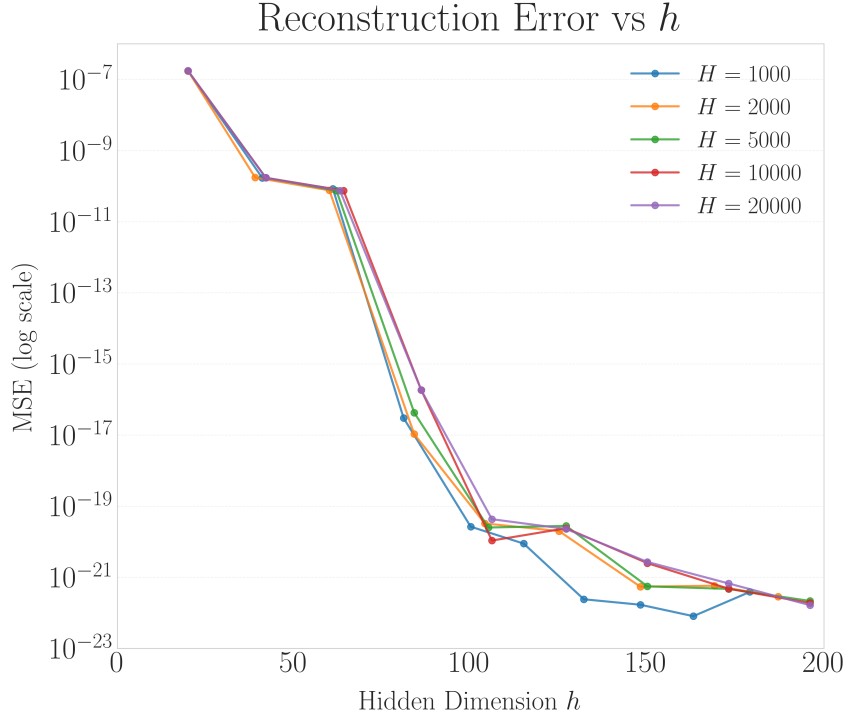

Figure 10: Reconstruction Error varying $H$ and LDS state dimension $h$.

### A.9 Reconstruction Error of the Spectral Filters with the Practical Algorithm and Uniform $\alpha$ Distribution

In Figure 11, we illustrate that if rather than choosing $(|\alpha_i| \stackrel{d}{=} 1 - U^4)$ with $U \sim \text{Unif}[0, 1]$ with an independent random sign, we instead choose $\alpha_i \in [-1, 1]$, we require larger $H$ to achieve similar performance. At sufficiently large $H$, uniform sampling will have sufficient coverage of $\alpha$ near 1 and $-1$. Thus, we instead prefer a distribution with a natural bias towards these extreme points, as $\alpha \in [-0.9, 0.9]$ decay towards 0 quickly.

### A.10 Experiments on Synthetic Tasks

We evaluate SpectraLDS (state dimension 160) against a strong baseline on a synthetic task, reporting results in Table 3 and Figure 12. A symmetric linear dynamical system with input dimension $d_{\text{in}}$ and state dimension $d_h$ is initialized by drawing all entries from a standard normal distribution, after which the update matrix $A$ is rescaled so its largest eigenvalue has magnitude $1 - \delta$. We then distill a SpectraLDS model from an STU model trained with AdaGrad (learning rate 1.0) for 2000 steps; each step minimizes the mean-squared-error (MSE) between the STU output and the ground-truth LDS output over batches of 32 sequences of length *seq_len*. For the baseline, we fit a randomly initialized symmetric LDS with AdaGrad (learning rate 0.0001) under the same loss. To keep training times practical, most baseline runs use a shorter sequence length, and several runs were stopped early due to computational limits, marked with an asterisk. **When the state dimension satisfies** $d_h \geq 1000$**, baseline training with a learning rate of** 0.01 **becomes unstable and often diverges, whereas the STU remains stable even with a learning rate of** 1.0**.**

Figure 11: Reconstruction Error varying $H$ and LDS state dimension $h$ when choosing $\alpha \in$ Unif$[-1, 1]$.

| Type | Len. | Delta | d_in | d_h | Avg MSE | MSE Std | Step 10 Loss | Step 100 Loss | Runs | Time (s) | Time Std (s) |
|------|------|-------|------|-----|---------|---------|--------------|---------------|------|----------|--------------|
| LDS GD | 8192 | $1 \times 10^{-2}$ | 10 | 100 | $2.15 \times 10^{-2}$ | $1.32 \times 10^{-3}$ | $3.49 \times 10^{-1}$ | $1.43 \times 10^{-1}$ | 4* | 2153.12 | 14.16 |
| LDS GD | 8192 | $1 \times 10^{-3}$ | 10 | 100 | $2.53 \times 10^{-2}$ | $3.18 \times 10^{-3}$ | $4.53 \times 10^{-1}$ | $1.80 \times 10^{-1}$ | 3* | 2147.61 | 18.64 |
| LDS GD | 8192 | $1 \times 10^{-4}$ | 10 | 100 | $3.03 \times 10^{-2}$ | $6.09 \times 10^{-3}$ | $5.99 \times 10^{-1}$ | $2.29 \times 10^{-1}$ | 3* | 2150.73 | 28.05 |
| LDS GD | 1024 | $1 \times 10^{-2}$ | 10 | 100 | $9.54 \times 10^{-3}$ | $0.00 \times 10^{0}$ | $1.18 \times 10^{-1}$ | $6.72 \times 10^{-2}$ | 1* | 248.33 | 0.00 |
| LDS GD | 1024 | $1 \times 10^{-2}$ | 10 | 1000 | $1.90 \times 10^{-5}$ | $5.93 \times 10^{-6}$ | $5.25 \times 10^{-4}$ | $3.39 \times 10^{-5}$ | 5 | 252.24 | 1.64 |
| LDS GD | 1024 | $1 \times 10^{-3}$ | 10 | 100 | $7.34 \times 10^{-3}$ | $1.02 \times 10^{-3}$ | $1.06 \times 10^{-1}$ | $4.97 \times 10^{-2}$ | 2* | 248.68 | 0.13 |
| LDS GD | 1024 | $1 \times 10^{-3}$ | 10 | 1000 | $2.40 \times 10^{-5}$ | $1.71 \times 10^{-6}$ | $4.00 \times 10^{-4}$ | $3.71 \times 10^{-5}$ | 5 | 250.24 | 0.87 |
| LDS GD | 1024 | $1 \times 10^{-4}$ | 10 | 100 | $7.72 \times 10^{-3}$ | $2.52 \times 10^{-4}$ | $1.13 \times 10^{-1}$ | $5.31 \times 10^{-2}$ | 2* | 248.32 | 0.38 |
| LDS GD | 1024 | $1 \times 10^{-4}$ | 10 | 1000 | $2.01 \times 10^{-4}$ | $4.53 \times 10^{-6}$ | $5.29 \times 10^{-4}$ | $3.53 \times 10^{-5}$ | 5 | 250.58 | 1.40 |
| LDS GD | 1024 | $1 \times 10^{-5}$ | 10 | 100 | $6.77 \times 10^{-3}$ | $6.43 \times 10^{-4}$ | $1.01 \times 10^{-1}$ | $4.84 \times 10^{-2}$ | 5 | 250.80 | 0.17 |
| LDS GD | 1024 | $1 \times 10^{-5}$ | 10 | 1000 | $1.94 \times 10^{-5}$ | $4.92 \times 10^{-6}$ | $4.79 \times 10^{-4}$ | $3.39 \times 10^{-5}$ | 5 | 251.51 | 0.67 |
| SpectraLDS | 8192 | $1 \times 10^{-2}$ | 10 | 100 | $3.35 \times 10^{-4}$ | $9.84 \times 10^{-5}$ | $8.27 \times 10^{-3}$ | $3.74 \times 10^{-4}$ | 5 | 51.85 | 10.36 |
| SpectraLDS | 8192 | $1 \times 10^{-2}$ | 10 | 1000 | $1.86 \times 10^{-5}$ | $2.27 \times 10^{-6}$ | $7.99 \times 10^{-3}$ | $5.48 \times 10^{-5}$ | 5 | 51.14 | 0.04 |
| SpectraLDS | 8192 | $1 \times 10^{-3}$ | 10 | 100 | $3.73 \times 10^{-4}$ | $1.57 \times 10^{-4}$ | $9.32 \times 10^{-3}$ | $4.12 \times 10^{-4}$ | 5 | 63.92 | 24.06 |
| SpectraLDS | 8192 | $1 \times 10^{-3}$ | 10 | 1000 | $1.73 \times 10^{-5}$ | $2.27 \times 10^{-6}$ | $8.46 \times 10^{-3}$ | $5.49 \times 10^{-5}$ | 5 | 51.93 | 1.55 |
| SpectraLDS | 8192 | $1 \times 10^{-4}$ | 10 | 100 | $4.73 \times 10^{-4}$ | $1.91 \times 10^{-4}$ | $9.81 \times 10^{-3}$ | $5.11 \times 10^{-4}$ | 5 | 52.23 | 10.66 |
| SpectraLDS | 8192 | $1 \times 10^{-4}$ | 10 | 1000 | $1.84 \times 10^{-5}$ | $4.53 \times 10^{-6}$ | $9.30 \times 10^{-3}$ | $5.54 \times 10^{-5}$ | 5 | 51.15 | 0.07 |
| SpectraLDS | 8192 | $1 \times 10^{-5}$ | 10 | 100 | $2.94 \times 10^{-4}$ | $8.78 \times 10^{-5}$ | $1.18 \times 10^{-2}$ | $3.30 \times 10^{-4}$ | 5 | 51.69 | 10.37 |
| SpectraLDS | 8192 | $1 \times 10^{-5}$ | 10 | 1000 | $1.66 \times 10^{-5}$ | $4.47 \times 10^{-6}$ | $9.00 \times 10^{-3}$ | $5.27 \times 10^{-5}$ | 5 | 51.15 | 0.07 |
| SpectraLDS | 8192 | $1 \times 10^{-6}$ | 10 | 100 | $4.05 \times 10^{-4}$ | $3.61 \times 10^{-5}$ | $1.07 \times 10^{-2}$ | $4.46 \times 10^{-4}$ | 5 | 46.53 | 0.03 |
| SpectraLDS | 8192 | $1 \times 10^{-6}$ | 10 | 1000 | $1.96 \times 10^{-5}$ | $1.27 \times 10^{-6}$ | $8.55 \times 10^{-3}$ | $5.63 \times 10^{-5}$ | 5 | 51.95 | 1.63 |
| SpectraLDS | 8192 | $1 \times 10^{-7}$ | 10 | 100 | $4.50 \times 10^{-4}$ | $1.73 \times 10^{-4}$ | $9.63 \times 10^{-3}$ | $4.87 \times 10^{-4}$ | 5 | 46.51 | 0.06 |
| SpectraLDS | 8192 | $1 \times 10^{-7}$ | 10 | 1000 | $1.77 \times 10^{-5}$ | $1.95 \times 10^{-6}$ | $9.45 \times 10^{-3}$ | $5.46 \times 10^{-5}$ | 5 | 51.96 | 1.64 |

Table 3: SpectraLDS performance on learning synthetic linear dynamical systems with maximum eigenvalue $1 - \delta$ against a strong baseline.

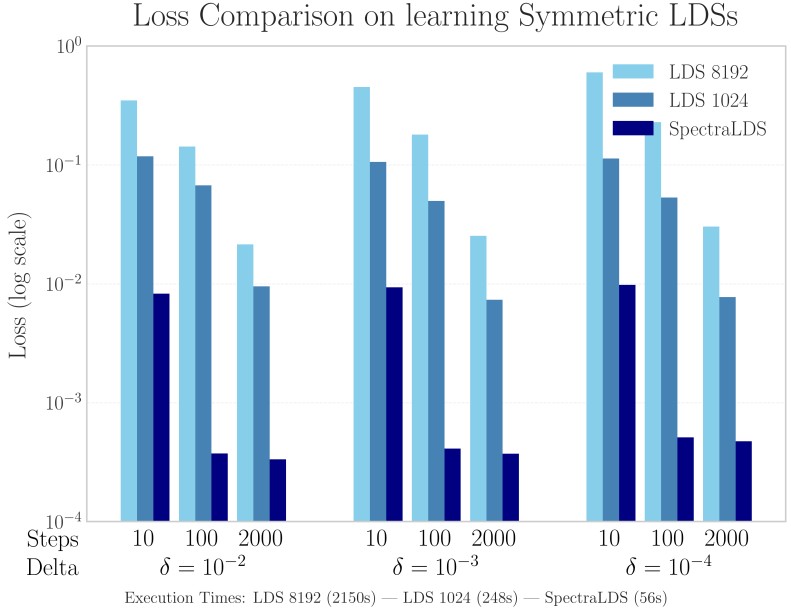

Figure 12: Comparison of learning performance on synthetic LDS tasks between SpectraLDS and a gradient-descent-updated LDS baseline. The plot shows the loss at different training steps under varying $\delta$, input/output dimensions, and sequence lengths. SpectraLDS consistently achieves lower loss with significantly reduced runtime, especially at larger output dimensions. Full experimental details are provided in Table 3.

### A.11 Layer Speeds of SpectraLDS and STU

We benchmark the inference speed of a single SpectraLDS layer across several state dimensions and compare it with two accelerated schemes for computing the STU convolution: Epoched Future Fill [1] and an STU that employs the tensor-dot approximation [22]. The SpectraLDS layer itself is produced by applying STU-to-LDS distillation to the tensor-dot STU. For the timings reported in Table 4, each model is evaluated on $L$ (Seq. Len.) autoregressive convolutions on inputs of dimension 128, and the mean and standard deviation over five runs are recorded. All STU variants use a filter length of 1,048,576 to accommodate the longest sequences. Although every architecture begins in the linear-scaling regime, only SpectraLDS continues to scale favorably as the sequence length increases. The resulting layer-speed comparison is visualized in Figure 13; benchmarks of SpectraLDS layers embedded in large language-model architectures appear in Appendix A.12.

| | Inference Time Performance (ms) | | | | | | | | | |
|---|---|---|---|---|---|---|---|---|---|---|
| **Seq. Len.** | **STU Future Fill** | | **STU Tensor-Dot** | | **STU** | | **SpectraLDS (SD 100)** | | **SpectraLDS (SD 800)** | |
| | **Mean** | **Std** | **Mean** | **Std** | **Mean** | **Std** | **Mean** | **Std** | **Mean** | **Std** |
| 32 | 30.86 | 1.20 | 31.87 | 6.43 | 134.96 | 22.13 | 21.27 | 3.86 | 18.98 | 0.73 |
| 64 | 36.52 | 0.72 | 28.92 | 0.42 | 123.16 | 2.54 | 22.40 | 0.14 | 22.27 | 0.29 |
| 128 | 49.87 | 0.52 | 40.20 | 6.63 | 135.94 | 2.28 | 29.49 | 0.18 | 29.72 | 0.58 |
| 256 | 75.73 | 0.35 | 45.98 | 1.12 | 162.74 | 2.90 | 43.54 | 0.27 | 43.62 | 0.47 |
| 512 | 136.67 | 7.90 | 70.24 | 0.61 | 211.66 | 2.99 | 71.40 | 0.56 | 73.01 | 0.88 |
| 1024 | 234.10 | 6.99 | 122.89 | 1.19 | 317.91 | 4.79 | 127.52 | 1.25 | 128.98 | 0.97 |
| 2048 | 441.09 | 3.69 | 226.47 | 8.02 | 525.65 | 5.09 | 241.16 | 4.43 | 252.58 | 8.78 |
| 4096 | 863.81 | 8.72 | 426.43 | 7.96 | 936.94 | 6.53 | 472.15 | 8.40 | 477.96 | 8.42 |
| 8192 | 1856.63 | 191.17 | 837.46 | 2.03 | 1772.30 | 20.46 | 921.55 | 10.23 | 929.06 | 9.76 |
| 16384 | 3377.29 | 33.16 | 1812.17 | 16.47 | 3434.19 | 30.83 | 1839.63 | 25.21 | 1846.04 | 17.43 |
| 32768 | 7139.35 | 515.40 | 4286.86 | 24.74 | 6721.83 | 19.65 | 3620.84 | 43.71 | 3686.90 | 26.45 |
| 65536 | 13485.25 | 234.22 | 11614.43 | 28.04 | 13478.53 | 89.72 | 7181.13 | 35.96 | 7362.73 | 78.96 |
| 131072 | 27252.83 | 240.34 | 36427.10 | 14.56 | 26703.77 | 323.97 | 14356.92 | 74.91 | 14649.56 | 162.32 |
| 262144 | 63437.26 | 123.25 | 117796.38 | 268.48 | 55775.49 | 275.16 | 28573.69 | 119.09 | 29156.42 | 113.55 |
| 524288 | 177502.89 | 651.96 | 365168.31 | 460.23 | 142896.29 | 621.72 | 57083.23 | 404.91 | 58027.04 | 276.39 |
| 1048576 | 576129.57 | 1425.12 | 1145862.83 | 1805.50 | 451607.11 | 721.73 | 114044.79 | 1215.07 | 115999.77 | 670.78 |

Table 4: Autoregressive Inference Time (ms) across model architectures (5 runs).

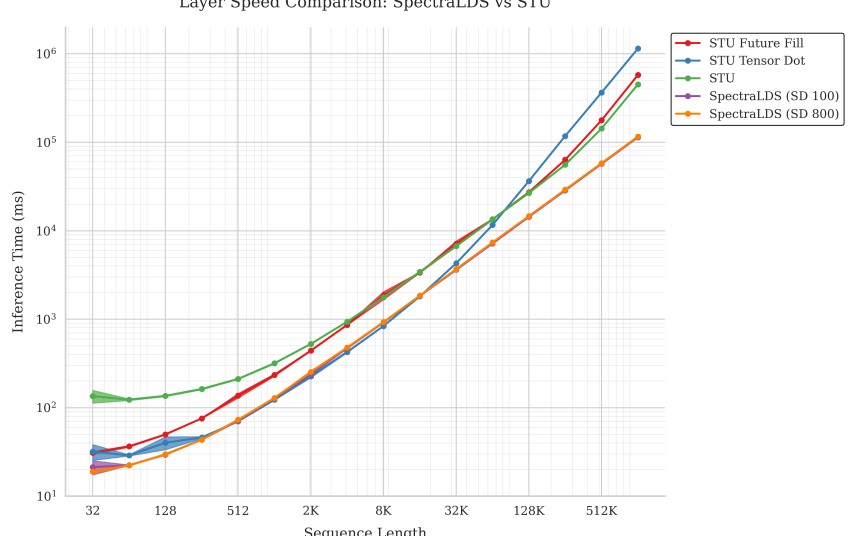

Figure 13: Autoregressive Inference Time (ms) across model architectures.

## A.12  FlashSTU Ablations

### A.12.1  Implementation Architecture Details

To perform the FlashSTU performance evaluations with STU-T, we employ the architecture depicted in Figure 14. We test token generation efficiency with a hybrid architecture, alternating between layers using STU-T and sliding window attention (SWA), and we additionally test with each layer using STU-T only. Each layer consists of RMS-Norm, followed by STU-T or SWA, followed by RMS-Norm, and then followed by an MLP. Inputs are tokenized by the o200k_base tokenizer and the FlashSTU model begins with an embedding layer, which is weight-tied to the output unembedding layer. To start generation, we add special tokens such as `<|endoftext|>` and `<|endofprompt|>`.

The sliding window attention layers leverage Flash Attention v2 [8, 7] and ALiBi position embeddings [35]. The tested model has input dimension 896 and 12 layers, which has 550.31 million parameters for the hybrid model, and 535.99 million parameters for the STU-only model. All layers are run in bfloat16 except for the LDS distilled layers, which require float64 precision. The Flash STU-T leverages the STU with tensor-dot [22] approximation rather than the base STU layer for faster inference, and thus we perform the STU-to-LDS distillation on the STU with tensor-dot approximation. For tests with generation length up to 131072, we leverage STU filter length of 131072. For the generation length of 262144, we leverage an STU filter length of 262144. For each runtime measurement, we first run a warmup generation, before reporting the mean of two generations of that length. All benchmarks include only inference time and not model setup or filter computation costs. Additionally, each MLP layer has hidden dimension $12\times$ the input dimension (MLP expansion factor). Other configuration details are identical to those in Table 7.

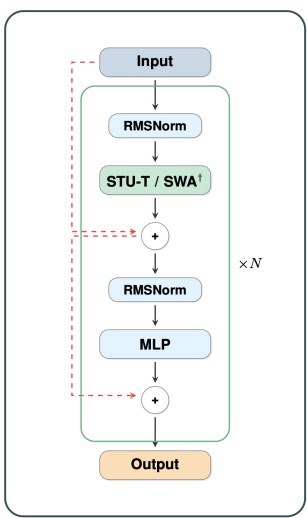

Figure 14: FlashSTU architecture [22].

### A.12.2 Implementation Efficiency

Using the setup of Fig. 14, we time autoregressive generation for sequence lengths 4,096–262,144 tokens under two architectures—(i) a hybrid network that interleaves STU-T and sliding-window attention layers and (ii) an STU-only network in which every layer is STU-T. Tables 5–6 report mean runtimes (over two runs after a warmup) for SpectraLDS with state dimensions 100–800 alongside the STU-T with naive convolutions (Base STU) and the Epoched FutureFill method. We note three main results. First, SpectraLDS runtimes grow nearly linearly with sequence length and are virtually independent of the chosen state dimension. Second, while the STU-T with naive convolutions is competitive at 4–8 k tokens, it becomes progressively slower, so that SpectraLDS is $\approx 2\times$ faster by 65 k tokens and over $4\times$ faster at 262 k tokens in the STU-only setting (and $2\times$ faster in the hybrid setting). Third, Epoched FutureFill narrows the gap at medium lengths but is still outpaced by SpectraLDS beyond 131 k tokens and, in the STU-Only architecture, exhausts memory (OOM) at 262 k tokens, whereas SpectraLDS completes the run. Together these results demonstrate that SpectraLDS delivers the most favorable long-context scaling and remains robust across model hyper-parameters.

| Seq. Len. | LDS SD 100 | LDS SD 200 | LDS SD 400 | LDS SD 800 | Base STU | FutureFill |
|---|---|---|---|---|---|---|
| 4096 | 20.34 | 20.71 | 20.40 | 20.40 | 18.91 | 19.92 |
| 8192 | 40.48 | 41.32 | 40.69 | 40.82 | 38.09 | 38.39 |
| 16384 | 80.78 | 82.90 | 80.95 | 81.63 | 76.49 | 74.89 |
| 32768 | 161.43 | 163.92 | 162.13 | 163.30 | 164.70 | 141.16 |
| 65536 | 323.40 | 327.31 | 323.90 | 325.92 | 389.72 | 290.68 |
| 131072 | 646.66 | 653.96 | 648.99 | 651.35 | 1014.67 | 666.24 |
| 262144 | 1588.81 | 1639.24 | 1591.50 | 1536.20 | 3113.45 | 2498.89 |

Table 5: Hybrid Model Runtime (seconds) for generation across SpectraLDS with different state dimensions and Baseline and Epoched FutureFill implementations

| Seq. Len. | LDS SD 100 | LDS SD 200 | LDS SD 400 | LDS SD 800 | Base STU | FutureFill |
|---|---|---|---|---|---|---|
| 4096 | 14.04 | 13.87 | 13.87 | 13.74 | 12.14 | 19.35 |
| 8192 | 28.15 | 27.74 | 27.67 | 27.42 | 24.84 | 35.86 |
| 16384 | 55.71 | 55.51 | 55.34 | 54.73 | 57.32 | 67.96 |
| 32768 | 110.85 | 110.88 | 110.57 | 108.68 | 149.12 | 135.75 |
| 65536 | 220.11 | 221.55 | 221.13 | 215.00 | 429.20 | 329.91 |
| 131072 | 439.87 | 442.65 | 442.04 | 428.36 | 1352.40 | 952.70 |
| 262144 | 1104.97 | 1101.76 | 1132.18 | 1107.49 | 4672.60 | OOM |

Table 6: STU-Only Runtime (seconds) for generation across SpectraLDS with different state dimensions and Baseline and Epoched FutureFill implementations

## A.13 Details of STU Model In Experiments

We summarize in Table 7 all relevant details for the FlashSTU model used in the language evaluations in Table 2. The distilled LDS layer used in the language benchmarking experiments was obtained by Algorithm 2 and has a state dimension of 160, incorporating both positive and negative spectral components. The weights for the distilled model were directly transferred from the FlashSTU model described below. The FlashSTU architecture is further described in Appendix A.12 and graphically shown in Figure 14.

Benchmarks were conducted with task-specific few-shot configurations. For each task, if the few-shot value is not set to -1, that number of examples is provided during evaluation; otherwise, the few-shot setting is left unchanged. Specifically, the few-shot settings are as follows: HellaSwag: 0 shots, MMLU: 5 shots, PIQA: 0 shots, BoolQ: 0 shots, Winogrande: -1, Commonsense_QA: 7 shots, OpenBookQA: -1, ARC-Easy: -1, ARC-Challenge: -1. The model employs bfloat16 precision except for the LDS computations, where it requires float64 precision, and a global batch size of 3 examples per task. All computations were performed on a single H100 GPU.

## A.14 Example Generation

The following is an example completion from the FlashSTU 340M model after LDS distillation with temperature 1.0:

**Prompt:** After school each day, Deidre Smith would look over the essays her students submitted at the end of class and, without fail, each day Alex would surprise her. Today, he wrote about

**Generated Text:** After school each day, Deidre Smith would look over the essays her students submitted at the end of class and, without fail, each day Alex would surprise her. Today, he wrote about 40 essays from his freshman class, all of which were collected over the past three years for The Book of Sentiments: What Students Should Do When Reading Literature – First published in 1962, this book was very well received. What people who read this book would not immediately accept or appreciate the author's thought-provoking message of what it means to read literature.

What exactly are you doing with your newfound ability to read? You are reading? What emotions do you feel at the time? What do you do with your newfound ability to read? These are just a few examples of how much reading students think they can gain from the experience. It is no accident, then, that each and every level of student will have difficulty with the text in this book. Students are given a choice: choose an assignment that you are comfortable with and one that you would like to read more deeply and critically. As the task to read becomes more difficult and the pressure for it is much greater, students will struggle to put their best minds together when reading literature. To help students overcome this hurdle, I have devised a book that makes the task a bit easier by providing all the information they need from a reader. If a reader is too afraid to read, I have designed a page that provides guidance from the textbook that will...

| Model Architecture | | |
|---|---|---|
| **** | **Description** | **Flash STU** |
| **Parameter Count** | Total number of parameters | 340M |
| **Embedding Dimension** | Dimensionality of embedding space | 1024 |
| **Number of Heads** | Attention heads | 4 |
| **Number of Layers** | Transformer + STU layers | 12 |
| **ALiBi Attention** | Attention scores modification using linear biases | No |
| **RoPE Theta** | RoPE scaling factor for rotary embeddings | 10,000 |
| **Sliding Window Size** | Sliding window attention context lookback size | 512 |
| **Sequence Length (Training)** | Input sequence length during training | 4,096 |
| **Sequence Length (Inference)** | Input sequence length during inference via position interpolation | 131,072 |
| **Vocabulary Size** | Size of the model's vocabulary | 200,064 |
| **MLP Expansion Factor** | Expansion factor in MLP layers | 4 |
| **Bias** | Use of bias terms in linear layers | No |
| **Dropout** | Dropout rate | 0.0 |
| **Number of Filters** | Number of filters | 24 |
| Training and Optimization | | |
| **Epochs** | Number of training epochs | 1 |
| **Global Batch Size** | Number of tokens processed per step | 524,288 |
| **Micro Batch Size** | Batch size per GPU | 8 |
| **Gradient Accumulation Steps** | Number of steps before performing a gradient update | 8 |
| **Warmup Steps** | Number of warmup steps | 1,907 |
| **Evaluation Period** | Evaluation frequency (steps) | 50 |
| **Max Grad Norm** | Maximum gradient norm for clipping | 1.0 |
| Optimizer Configuration | | |
| **Optimizer** | Optimizer type | AdamW |
| **Learning Rate Schedule** | LR scheduling strategy | Linear decay with warmup |
| **Max Learning Rate** | Maximum learning rate | $4.0 \times 10^{-4}$ |
| **Min Learning Rate** | Minimum learning rate | $4.0 \times 10^{-5}$ |
| **Torch Dtype** | Data type for PyTorch tensors | `bfloat16` |
| **Betas** | Optimizer betas | (0.9, 0.999) |
| **Epsilon** | Optimizer epsilon | $1.0 \times 10^{-8}$ |
| **Weight Decay** | Weight decay factor | $1.0 \times 10^{-2}$ |
| **AMSGrad** | Use AMSGrad variant | No |
| **Fused** | Use fused optimizer | Yes |
| Optimization Techniques | | |
| **Activation Checkpointing** | Enable activation checkpointing | Yes |
| **Use Flash FFT** | Enable Flash FFT | No |
| **Use Tensordot Approx.** | Enable tensordot approximation | Yes |
| **Use Attention** | Enable attention mechanism | Yes |
| **Softcap** | Softcap threshold | 50.0 |
| **Torch Compile** | Enable Torch compile optimization | Yes |

Table 7: Model and training configuration details for the 340M Flash STU model.

