# OpenReview forum: "SpectraLDS: Provable Distillation for Linear Dynamical Systems"
_NeurIPS.cc/2025/Conference — NeurIPS 2025 poster_

### Official Review · Reviewer_TKYC · 2025-07-01

**Clarity:** 3
**Significance:** 1
**Originality:** 1
**Rating:** 3
**Confidence:** 3

**Summary:**

The authors present a method to distill spectral filter-based models (STUs) into state-space model-based (linear dynamical systems, LDS) representations. They provide a provable technique for this distillation that guarantees accuracy independent of the system’s state dimension or effective memory. Empirical evidence is presented demonstrating that their approach maintains predictive accuracy while substantially improving inference efficiency.

**Questions:**

Can the authors clarify or better motivate the practical significance or broader implications of this distillation technique?

**Ethical Concerns:**

["NO or VERY MINOR ethics concerns only"]

**Final Justification:**

I appreciate the authors' detailed rebuttal and engagement.

The authors acknowledge the significant restriction of their approach to symmetric transition matrices. In my view, this restriction considerably weakens the contribution, as filters from symmetric LDSs are essentially limited to linear combinations of exponential decays, forming a low-dimensional representation space. Consequently, I find it unsurprising that such filters can be effectively approximated by randomly sampling diagonal real LDS elements and linearly combining them.

Nevertheless, I acknowledge that within the broader context of literature addressing the challenge of distilling diverse filter representations into LDS frameworks, this result holds some value. Given this recognition, I have revised my score upward to a 3.

**Limitations:**

yes.

**Paper Formatting Concerns:**

None.

**Quality:**

3

**Strengths And Weaknesses:**

**Strengths**:
1. Well-written: The paper clearly explains it's contribution, methorodlogy, and experiments.
2. Thoroughness: The authors include extensive theoretical analysis, proofs, and multiple experimental validations, complemented by numerous detailed expansions in the appendix.
3. Convincing experiments: The experiments robustly validate their claims, showing that the distillation effectively maintains accuracy across multiple benchmarks.

**Weaknesses**:
1. Although somewhat subjective, the result appears to be somewhat niche and not strongly motivated. Specifically, STUs are not currently predominant in sequence modeling, and it is unclear whether they significantly outperform existing SSM-based models, many of which already achieve constant-time and constant-space inference per token. Even if in certain settings STUs do outperform other approaches, writing an entire paper solely about distilling such models into LDS form seems like too narrow of a contribution.
2. Again, subjective, but I find that both the proposed algorithm and associated theoretical proofs are rather simple (and not surprising), raising the question of whether the overall contribution is substantial enough to warrant acceptance. Specifically, the main result (Algorithm 2) simply randomly samples a diagonal LDS and linearly fits its parameters to match the spectral filters; this seems to me more like a straightforward linear approximation of spectral filters using randomly chosen exponential bases rather than a deep or novel theoretical insight.

---

> ### Author Rebuttal · Authors · 2025-07-30
>
> Dear Reviewer,
>
> Thank you for your time and your feedback. We appreciate your concerns about the scope of our results, especially as the STU is not as popular as other SSM architectures. However, our work carries large implications for the LDS, a common primitive forming the basis of most SSM architectures, and whose pitfalls provide intuition for many popular SSMs, such as S4 [10], Hyena [32], and Mamba [9].
>
> Regarding your concern about simplicity: we respectfully disagree. We argue that simplicity is an advantage. Our method, while straightforward, is surprising, and circumvents the long-held understanding of parameter fitting of an LDS. Decades of work has culminated in methods that work in special cases (no noise or normally distributed noise), or agnostic improper learning (online) via convex relaxation. The new development of spectral filtering allows our proper learning of an LDS via learning the filters, rather than directly, which we consider unexpected. It is also worth noting that, although in our main result we present the more straightforward algorithm, this algorithm is far from the most practically performant, and we introduce more complex algorithms with substantially better performance in the appendix.
>
> As far as the broader impact of our work, historically, despite their theoretical maturity, there have been no efficient and provably accurate learning methods for fitting LDS parameters to a signal with guarantees independent of the effective-memory of the system. This challenge has been a popular area of research over the last few years, with the Legendre Memory Unit [1*] and HiPPO initialization [2*] being significant results in this line of work. Although our result is currently restricted to fitting LDSs with symmetric transition matrices, we provide what is, to our knowledge, the strongest result to date for learning LDSs with high effective-memory, with performance independent of the effective-memory the signal and with strong convergence bounds.
>
> Below, we will mention a few other examples of the practical significance of our work:
>
> The Laughing Hyena Distillery (LHD) [26] technique has facilitated convolutional models 10x faster than Transformers (at equal performance) for language modeling. LHD transforms convolutional layers into complex symmetric LDS layers with gradient methods, offering no guarantees but having strong practical performance. With SpectraLDS, we can now convert convolutional layers into a real symmetric LDS without attempting non-convex optimization by projecting onto the spectral basis. Moreover, we also now have an understanding of which filters can be well represented, and the SpectraLDS layer serves as a drop-in method for learning ‘fast’ convolutional layers. As prior work has shown with the STU [2], the layer has high learning capacity on sequence-to-sequence tasks, as it is well suited for oscillating or decaying dependence.
>
> For a signal produced by a symmetric LDS, we have now provided a strong bound for how increasing LDS parameters corresponds to reducing error. As a corollary of our main result, we have that, with a bounded input sequence, loss is bounded above by $O(e^{-p})$ when fitting an LDS with SpectraLDS, where p is the number of parameters and the constant is well understood. To our knowledge, this is the strongest scaling law for an LDS. As prior work and industry practice has shown success in modeling audio, among other signals, with increasingly large LDS models, the knowledge of scaling remains important. To be clear, our bound is still weaker than what is practically observed, but we still believe this is an important step in the right direction.
>
> Moreover, recent work [3*] has extended the STU to a larger class of LDSs, with performance dependent on the maximal imaginary component of the eigenvalue of the transition matrix. In this case, our work may have applications to a much larger class of LDSs than we had initially anticipated, and provide the basis for general LDS training and understanding.
>
> We appreciate your feedback and thank you for sharing your honest opinions. We hope we have largely addressed your concerns, and we would be happy for you to elaborate on any concerns that you believe still remain in the paper. In our revision, we will improve the discussion section to make the impact more visible, and we will include some of the points we have raised here.
>
> Thank you for your time and feedback,
>
> Authors
>
> [1*] Voelker, A., Kajić, I., & Eliasmith, C. (2019). Legendre Memory Units: Continuous-Time Representation in Recurrent Neural Networks. In H. Wallach, H. Larochelle, A. Beygelzimer, F. d'Alché-Buc, E. Fox, & R. Garnett (Eds.), Advances in Neural Information Processing Systems (Vol. 32).
>
> [2*] Gu, A., Dao, T., Ermon, S., Rudra, A., & Ré, C. (2020). HiPPO: Recurrent Memory with Optimal Polynomial Projections. In H. Larochelle, M. Ranzato, R. Hadsell, M. F. Balcan, & H. Lin (Eds.), Advances in Neural Information Processing Systems (Vol. 33, pp. 1474–1487).
>
> [3*] Marsden, A., & Hazan, E. (2025). Universal Sequence Preconditioning. arXiv.

---

> > ### Comment · Reviewer_TKYC · 2025-08-02
> >
> > I appreciate the authors' detailed rebuttal and engagement.
> >
> > The authors acknowledge a significant restriction:
> >
> > > "Although our result is currently restricted to fitting LDSs with symmetric transition matrices."
> >
> > *I admit that this is a subjective matter*, but in my view, this restriction considerably weakens the paper's contribution. Filters generated by LDSs with symmetric transition matrices can be diagonalized into LDSs with diagonal and real transition matrices and thus represented as linear combinations of exponential decays. Therefore, the space of filters produced by symmetric transition matrices is essentially limited to linear combinations of exponential decays, effectively residing in a very low-dimensional space (as previously shown in the original STU paper). Consequently, I find it unsurprising that any filter generated by a linear LDS can be approximated effectively by randomly "sampling" elements of a diagonal real LDS (i.e., sampling exponential decays) and fitting B (linear combination of these exponential decays) to match the filter. In my opinion, this constitutes a first-cut technique, and the proof to this first-cut technique is simple.
> >
> > *Before locking my score, I would like to give the authors an opportunity to correct me if my interpretation of the algorithm is incorrect, or if my intuition regarding its correctness is overly simplified. I'd appreciate it if their response could focus specifically on clarifying this aspect rather than debating the subjective question of whether this weakens the contribution.*
> >
> > While I still think that the restriction to symmetric transition matrices reduces the contribution too significantly to warrant acceptance, I do acknowledge that this result is somewhat valuable now seen within the  broader literature focused on the general challenge of effectively distilling diverse filter representations into LDS frameworks. **Given this recognition, I will increase my score to a 3.**

---

> ### Author Response · Authors · 2025-08-05
>
> We appreciate the reviewer’s careful engagement and are grateful for the opportunity to clarify this point.
>
> .
>
> *On the expressivity of symmetric LDSs:*
>
> It is correct that symmetric transition matrices yield real eigenvalues, and thus the impulse responses of such systems can be decomposed into linear combinations of exponential decays -- i.e., $(\rho^t)$ for $|\rho| < 1$. However, this space includes alternating sequences such as $f(t)=(−0.99)^t$, which are qualitatively different from smooth monotonic decays and can be difficult to capture with parameterized convolutional kernels.
>
> These alternating sequences—common in modalities with an intrinsic sampling rate such as audio— are nontrivial to approximate and are challenging to express accurately with many kernel parameterizations. Notably, these eigenvalues are also the most challenging to learn with most deep learning methods but necessary to learn expressive symmetric (and asymmetric) LDSs.
>
> While our theory applies to LDSs with symmetric transition matrices, we tested our method on filtered generated by LDSs with asymmetric transition matrices to demonstrate that the method generalizes well in practice and performs well on general LDSs.
>
> We performed a series of experiments that tested the capacity of SpectraLDS to learn an LDS with asymmetric transition matrix and high effective memory compared with baselines of fitting an (asymmetric / general) LDS with the same state dimension and fitting an LDS with twice the state dimension. For both the baselines, to improve training stability, we set a learning rate of 1e-3, and we initialize the models with spectral norm of A bounded by 1. For the STU, we use learning rate 1.0. For all models, we use the Adagrad optimizer and gradient clipping.
>
> In our experiments we found that, despite their stability adjustments, both baselines converged slower, attained worse fits, and remained unstable. In contrast, SpectraLDS attained better results substantially faster and with greater stability. It is worth noting that SpectraLDS almost always reaches near-convergence within 100 steps, and in the first 3 configurations, reached the same loss at the baselines more than 60x as quickly. We’ve included tables summarizing our results in a sub-comment.
>
> .
>
> *On the surprisingness of the approximation via exponential sampling.*
>
> We respectfully disagree with the reviewer. We would like to offer some additional context that we hope will clarify why we view the outcome as more surprising and significant than it may initially seem.
>
> While the final algorithm is relatively short, arriving at this approach was not straightforward. In practice, many alternative strategies that may appear promising fail to work. For example, in early attempts to distill the FlashSTU model [22], we trained both symmetric and asymmetric dynamical systems directly to fit the STU layers, a task analogous to learning a symmetric LDS. Despite using models with state dimensions exceeding 80,000, we found that these approaches only captured the signal accurately over the most recent 50 inputs (and incurred a substantial cost in efficiency). These difficulties occurred despite careful initialization of A and considerable effort to ensure stable training.
>
> We used established methods for training such systems and found that, despite a substantial body of literature on fitting symmetric or near-symmetric LDSs, existing techniques struggled significantly in this setting and had not focused on sufficiently long-horizon tasks. This motivated the development of our approach.
>
> We believe it surprising that the best way to fit an LDS involves first projecting to the spectral basis of Hankel eigenvectors, before then fitting each of the spectral filters with an LDS, In contrast, directly fitting the LDS without this transformation—even with significantly more parameters—was far less effective. Our experiments indicate that the spectral projection acts as a denoising step, which simplifies the learning problem and improves both stability and accuracy, but had not yet been fully understood.
>
> In retrospect it seems simple, but to the extent of our testing, this is the only algorithm that was properly able to learn the LDS intrinsic to the FlashSTU. Moreover, the fact that a relatively short algorithm can be tuned to have great practical performance is part of what we believe will make this work more valuable.
>
> With Algorithm 3, we have found a robust technique that is practically performant and well understood, and allows for the production of convolutional layers that are easily accelerated at inference.
>
> Much of the surprise of our work comes from the result that our technique works well practically. In all tests we have performed, SpectraLDS has performed much better than the next best alternative, and we are excited to be able to bring such an algorithm to the broader community.

---

> ### Author Response · Authors · 2025-08-05
>
> We apologize for the double comment, the experiment table expends many characters in the response.
>
> ---
>
> *Experiment Results:*
> The configs are in order of difficulty and are:
>
> | Name     | \$d\_h\$ | \$d\_{in}/d\_{out}\$ | Spectral Radius | Seq Len | Steps |
> | -------- | -------- | -------------------- | --------------- | ------- | ----- |
> | Config 1 | 50       | 50                   | 0.90            | 500     | 1500  |
> | Config 2 | 100      | 100                  | 0.95            | 1000    | 2000  |
> | Config 3 | 200      | 100                  | 0.99            | 2000    | 2500  |
> | Config 4 | 150      | 150                  | 0.999           | 1500    | 3000  |
>
> To provide a fair comparison, average losses and time are reported only for runs that converged (final loss \$\leq 100\$). All tests were run on an H100-80GB Nvidia GPU.
>
> | Config | Model        | Runs | Converged % | Avg Final Loss | Avg Total Time (s) | Avg Loss at 100 Steps |
> | ------ | ------------ | ---- | ----------- | -------------- | ------------------ | --------------------- |
> | 1      | STU          | 10   | 100.0%      | 0.126684       | 39.97              | 0.181331              |
> | 1      | BASELINE     | 10   | 100.0%      | 2.155365       | 163.99             | 5.345679              |
> | 1      | BASELINE\_2X | 10   | 100.0%      | 0.703029       | 164.52             | 4.425588              |
> | 2      | STU          | 10   | 100.0%      | 0.437770       | 120.54             | 0.545244              |
> | 2      | BASELINE     | 10   | 100.0%      | 1.987184       | 438.23             | 5.780517              |
> | 2      | BASELINE\_2X | 10   | 90.0%       | 0.690255       | 446.48             | 4.886963              |
> | 3      | STU          | 5    | 100.0%      | 6.330055       | 318.47             | 6.922692              |
> | 3      | BASELINE     | 5    | 60.0%       | 7.932160       | 1145.82            | 12.150135             |
> | 3      | BASELINE\_2X | 5    | 0.0%        | N/A            | N/A                | N/A                   |
> | 4      | STU          | 3    | 100.0%      | 20.189109      | 529.73             | 21.401643             |
> | 4      | BASELINE     | 3    | 66.7%       | 19.439209      | 1308.05            | 30.539234             |
> | 4      | BASELINE\_2X | 3    | 0.0%        | N/A            | N/A                | N/A                   |
>
> ---
>
> Across these experiments, the STU achieved near convergence within 100 steps in almost all cases and outperformed baselines by a wide margin, both in accuracy and runtime. The mean error increase across STU-to-SpectraLDS conversions was 10% (min: 1.8%, max: 27%), thus preserving the considerable benefits of training speed and stability. In this experiment, we chose a SpectraLDS with hidden dimension 160, although note this does not affect STU training. A larger hidden dimension would reduce the conversion error.
>
> We emphasize that SpectraLDS can be further refined into an asymmetric model for further improvement (at the cost of greater inference cost) if desired.
>
> .
>
>
> *Summary*
>
> We are grateful for the thoughtful engagement of the reviewer and for offering us the opportunity to clarify our work. We agree that there are multiple places we can improve the exposition of our work and the discussion of its implications, and we will do so in the camera-ready copy.
>
> While the restriction to symmetric LDSs is a fair concern, we hope that we have shown that even within this space, there are meaningful challenges and that our method offers a surprisingly effective solution. We are excited by how well it works in practice, for both symmetric matrices and asymmetric matrices, and we would be more than happy to run additional experiments to test other aspects. We also note that such results fall in line with prior work (e.g., S4 \[10]), and imposing symmetric and near-symmetric structure on the transition matrix has not prevented other architectures from SoTA attention-free performance.
>
> Thank you for a valuable conversation and helping strengthen our paper,
>
> Authors

---

> > ### Comment · Reviewer_TKYC · 2025-08-05
> >
> > Thank you for the meaningful discussion.
> >
> > I fully acknowledge that many findings may appear unsurprising or trivial in hindsight, and it is indeed challenging for reviewers to judge the complexity or subtlety of certain results accurately. **If it were possible, I would lower my confidence score to a 3, or even a 2.5.**
> >
> > Nevertheless, as an author, my responsibility is to assess the novelty and significance of the results to the best of my ability. Even upon reflection, I personally find the fact that LDS generated by symmetric systems can be effectively represented by simply sampling elements on the diagonal unsuprizing. Thus, I have chosen to maintain my original score.
> >
> > I appreciate your insightful comments and fully understand that my weaker points rely on subjective judgment, making them inherently more difficult to convincingly address.

---

### Official Review · Reviewer_ssg6 · 2025-07-01

**Clarity:** 2
**Significance:** 3
**Originality:** 3
**Rating:** 5
**Confidence:** 3

**Summary:**

The paper introduces an algorithm to distill a neural network module Spectral Transform Unit (STU) into a standard state space model (or control-affine linear system).  STU implicitly models state space models with real eigenvalues by learning a basis of spectral filters, which in turn define the coefficients of linear combinations of the input tokens, thus mimicking a linear attention mechanism. The objective is to improve the token-generation computational complexity of STUs without losing their expressivity.

The STU hinges on the fundamental assumption that the linear dynamics matrix of a control-affine linear system is symmetric and hence constrained to be diagonalizable and to have real eigenvalues. This assumption is required to satisfy the strong decay of the magnitude of eigenvalues of the Hankel matrix characterising the time evolution of the system’s linear dynamics matrix, which is the main argument to motivate the approximation of the most relevant spectral filters—the top eigenvectors of this Hankel matrix.

The proposed methodology takes as an input an STU with learned top spectral filters (eigenvectors of the Hankel matrix) and tries to estimate the parameters of a control-affine linear system that would lead to such spectral filters. The authors claim that this distillation process can be achieved rigorously, selecting the distilled state dimension as a function fo the approximation error of the STU.

**Questions:**

- The main suggestion is to enlarge the presentation and details of the proposed distillation approach in the main text. Specifically, I am concerned on the reproducibility of this distillation process by others, and the number of hyperparameters and intermediate steps omitted to be discussed in the main text. I would suggest to have a detailed version of the algorithm (like Algorithm 3) in the main text.

- Furthermore, I would like authors to clarify in detail in the main text:
    - How the dataset of 1D-LDS and corresponding STUs spectral filters, and spectral filter expansion coefficients of the LDS time evolution, is generated and used. Is this process completedly done offline and is it independent on the STU we are trying to distill?
    - How many iterations of process of sampling a subset of this dataset are used in practice to determine the initial subset of the dataset used?.
    - How to determine when to stop the “greedy” expansion of this dataset? Is this process determining the final dimension of the distilled LDS?
    - How is the final state dimension of the distilled LDS chosen?.

**Ethical Concerns:**

["NO or VERY MINOR ethics concerns only"]

**Final Justification:**

Based on the authors' rebuttal, I believe that they can successfully revise manuscript to integrate all the remarks from myself and other reviewers and meet the acceptance bar.  Therefore, I rise my score. That said, I find that revision is very much needed to improve clarity, especially issues of limiting to symmetric LDS.

**Limitations:**

Limitations were not discussed at all. While other aspects should be addressed, too, I believe it is important to elaborate on how prohibitive  the assumption of real-eigenvalues (symmetric LDSs) can be in practice. Personally, I am unaware if this assumption is reasonable / prohibiting in the context of language modelling. Authors could provide a brief statement on the implications of this assumption, or to references that discuss this topic.

**Quality:**

3

**Strengths And Weaknesses:**

### __Strengths__

- The general motivation of the paper is clear and well-articulated once the reader is familiar with STUs. The goal is to improve computational complexity of token generation in generative sequential models for language modeling.
- The different types of generative sequential model architectures and their respective computational complexity are well presented and explained, with Table 1 providing easy access to this information.
- Related work comparison is well structured and complete. I specially appreciated the references to standard literature on system identification.
- Experimental evaluations appear to be comprehensive and support the main thesis of the paper—that a distillation of an STU into a LDS($C$, $A$,$B$) with diagonal $A$,$C$, and $B$ can be achieved without losing precision while gaining substantial improvements in inference computational complexity.
- Once the reader is familiar with the concept of an STU, the paper's core idea and motivation are clearly explained and well-justified.

### __Weaknesses__

- __[Major]__ The main contribution of the paper is a procedure to distill a STU into a linear control affine form. Hence to comprehend and understand the motivation of the paper, it is crucial that the reader is aware of what an STU is, or that a good intuitive idea of an STU is presented to the reader early on. Unfourtanetly, as a reader unfamiliar with spectral attention methods, I was forced to reading references [2] and [14] to get a good intuitive and technical idea of an STU.

While producing self-contained theoretical research papers is challenging, I believe the authors could better introduce STUs and their relationship to SSMs in the introduction. This would help the average ML reader better understand and appreciate the work. Note that the terminology "spectral transform unit," "spectral filtering method," "spectral filter," and "impulse response" in the introduction provide little-to-no intuition to the average reader, as these terms aren't widely known in the community—or are used in non-standard ways.

Although STUs and spectral filtering methods are difficult to summarize intuitively in few words, I believe that the authors should make an effort to present these concepts clearly. Moving Fig. 2 to the introduction might help by showing both a standard evolution of an eigenfunction (what you call "impulse response") and the basis of spectral filters used to express the impulse response as a linear combination. This closely relates to the paper's objective. An early introduction of the transition from spectral filters to eigenvalues—with supporting visualisations—would dramatically improve the paper's accessibility.

- __[Major]__ The core contribution of the paper, explained in Section 5, is restricted to a single page, leaving several of the most important details of the proposed methodology unclear. For instance I was confused with:
    - Algorithm 1. appears to propose to solve a linear regression of the time evolution of an eigenfunction of $A$ (called ”impulse response” in the paper) in the basis of the spectral filters. This represents simply a linear transformation on the time (sequence-length) dimension, changing basis from time (/token order) to a spectral basis, in the same spirit as in a Fourier transform. It is unclear to me why there is a need for generating a random input vector, as an intermediate step to solve this process.
    - Algorithm 2. Line 3. States that to learn the basis expansion coefficients a sample of $h$ random eigenvalues $A$ is needed. However in the appendix it is explicitly stated that the STU→LDS distillation hinges on a dataset of $h$ synthetic 1-dimensional LDS systems with parameters ($\alpha_i$,, $b_i$, $c_i$), where the three scalar parameters are randomly sampled. It is unclear why algorithm 2 does not mention the generation of this datasets of 1-dimensional LDSs or why the sampling of the $B$ and $C$ parameters is ignored.
    - Algorithm 3 and appendix A.5 mention a "suitably chosen subset" of 1D-LDS systems (and their STU's representation) used to train the distillation process. This selection process of subsets and "greedy expansion of the subset" appears to be a hand-engineered step of the algorithm that is not well explained in Algorithm 3/appendix A.5, and is crucially omitted entirely from the main body.
    - Appendix A.5.4 mentions a fine-tunning of the approximated matrix M using gradient descent. This subprocess of the algorithm is again completedly omited in the main text.
- __[Minor]__ The expansion after line 150 assumes the state and noise a time zero are zero vectors, but this is never stated.
- __[Minor]__ In the equation after line 207, what was called previously an “impulse response”  $(\mu_{L}(\alpha)$ is now called a “single spectral filter” and conflated with the previously defined spectral filters $\phi_k$.
- __[Minor]__ Confusing nomenclature.
    - The term “impulse response” in control theory is associated with the response of a controlled dynamical system to an step function on its control signal. Given that there is an overlap between control and machine learning nomenclature, the use of the term “impulse response” to denote the vector of discrete-time evolution of an eigefunction of a dynamical system is confising.
    - Linear Dynamical Systems (LDS). For machine learning and control theoretic communities, LDS hints at a uncontrolled linear dynamical system, not a control-affine linear system, or a state space model.
    - It is unclear what the readers imply by “LDS filters” (line 205)
    - Line 3 of algorithm 2 says: “Sample h randomly chosen independent vectors $\alpha_1$…”, but \alpha is used to denote (scalar) eigenvalues.
- __[Minor]__ Typo or wrongly format sentence in line 57. Probably need to remove “this”.

---

> ### Author Rebuttal · Authors · 2025-07-30
>
> Dear Reviewer,
>
> Thank you for your thorough and constructive feedback, and we appreciate the abundance of detail! Below, we will address each point.
>
>
>
> .
>
>
> **[Major]** As a reader unfamiliar with spectral attention methods, I was forced to read references [2] and [14] to get a good intuitive and technical idea of an STU.
>
> We appreciate the extra time you spent on understanding our work, and we agree this certainly should not be necessary. Since submission, we have tried hard to improve accessibility and clarify core intuition, as well as reduce jargon in places we can. This has involved paper edits, but also the preparation of accompanying blog posts and content. For the submitted paper, we had added a section on notation in Appendix A.1, where we introduce the “impulse response” notation, among others, but in our revision we plan to add a formal glossary so the interested reader can accelerate through the introduction, and if they miss a definition, can easily glance to check.
>
> We agree largely with the reviewer that more visuals would be helpful, and we plan to move Figure 2 to the introduction as you suggested, and we also plan to add Fig. 2 from [2], which is a convenient schematic for showing how the STU operates by decomposing an input filter into the spectral basis.
>
> .
>
>
>
> **[Major]** The core contribution of the paper, explained in Section 5, is restricted to a single page, leaving several of the most important details of the proposed methodology unclear.
>
> **Algorithm 1:** Your assessment is correct, the random input vector is not necessary as an intermediate and the convolution suffices as well. We chose to introduce Algorithm 1 in this manner as we thought it was simpler to digest, and it has the same performance both theoretically and in practice, and both approaches are exceptionally quick.
>
> **Algorithm 2:** For ease of reading, we have switched all notation to the first version of sampling $h$ random eigenvalues of $A$. However, both sections are accurate as the b and c coefficients can be incorporated into the matrix M, and training with them randomly sampled or with b = c = 1 are both very effective.
>
> **Algorithm 3 and appendix A.5:** Thank you for the observation, we will be adding a table with all the parameters used for this part of the algorithm. In our forthcoming open-source release, we will provide code and all files needed for a complete reproduction, including all parameters for our results. We want to ensure that all aspects of this work are reproducible and publicly accessible to build on, and we thank the review for noticing areas we fell short on. We will additionally publish all model weights, infrastructure, and our set of 61,784 LDS-STU pairs used in Algorithm 3.
>
> **Appendix A.5.4:** After the matrix inversion, we ran 50,000 steps of gradient descent to improve the fit of the coefficients, although this was a relatively minor improvement compared to the rest of the algorithm. We omitted this in the main text for clarity, although it will be included in our open-source filter generation code and in our revised section 5.2.
>
> Generally, we agree the main section was not as clear as we would like, and we plan to address this by clarifying the above in the main paper, and by including the revised section 5.2, which we will discuss below.
>
>
> .
>
>
>
> **[Minor]**
> Thank you for catching the errors on lines 57, 150, 207, and in algorithm 2. We will fix them in the revision.
>
> Thank you for the suggestion with the “impulse response.” We will add a clear definition and glossary reference early on to avoid any misconception.
>
> The LDS filters refer to the convolutional kernel corresponding to the LDS, but we will define this more formally earlier.
>
> Thank you for the tip on the LDS definition. We have found the definition remarkably inconsistent across SSM papers, and we will be more formal in our earlier definitions to make this point clear.
>
>
> .
>
>
> **[Questions]**
>
> We certainly agree that it is important we improve the presentation and build greater STU intuition at the start. The additions of the new figures earlier on should hopefully help align the reader better with the STU and prepare them for our discussion.
>
> Additionally, we plan to improve exposition of our main algorithm and clarify the additional steps taken in the practical algorithm. In earlier drafts of the paper, we had Algorithm 3 in the main paper. However, this made the exposition substantially more challenging and we felt the core ideas were less accessible. We plan to move section 5.2 to the appendix and instead replace it with a short text description of the improvements in Algorithm 3, leaving the full description of Algorithm 3, including all of our hyperparameters, to the appendix. We have included an example of the revised section 5.2 at the end of our response.
>
> Generating the dataset of 1D-LDSs and corresponding filters is done offline and only needs to be done once as long as the STUs use the same set of spectral filters. We had fit 61,784 pairs offline, as described in section A.10, but the algorithm still has strong performance well below 10,000 pairs. In our forthcoming open-source release, we will provide the pair set we have computed. We will add full details for all the parameter choices in Algorithm 3 elsewhere in the appendix, and we will move a few of the core details, such as improving distribution election for $\alpha$, the gradient descent improvement, and the techniques to optimally select the eigenvalues of A to the main paper. Our hope is to strike a balance between being complete in the main paper while avoiding cluttering the main section and losing readability.
>
> The final state dimension was chosen by examining filter plots, such as Figure 7, and reconstruction losses, such as Figure 9, and finding that the sharpest loss drop is in the first 60 state dimensions. The state dimension of 80 was validated in our testing, but it is possible larger models may need up to 160 state-dimension and smaller models can use much less.
>
> .
>
>
> **[Limitations]**
>
> We agree that the restriction to symmetric LDSs is important to understand. This aspect was largely studied in earlier work on the STU [2], with the general results that although asymmetric LDSs have greater modeling capacity, they are practically bottlenecked on high-effective memory tasks by the challenges of training. In the S4 paper [10], which studied an expanded class of LDSs modeled by a normal plus low-rank matrix, the authors found that the LDSs they learned are nearly symmetric.
>
> Recent work [1*] has extended the STU to a larger class of LDSs, with performance dependent on the maximal imaginary component of the eigenvalues of the transition matrix. In this case, our work may have applications to a much larger class of LDSs than we had initially anticipated.
>
>
> .
>
> **[Closing]**
>
> We would be more than happy to elaborate on any lingering concerns or issues we have not yet suitably addressed. As a larger comment on the concerns of missing certain parameters, we care deeply about making this work something that people can replicate and build on, and we appreciate your feedback on cases where we fall slightly short. We have a full open-source release prepared, and we will be releasing all code, checkpoints, and data that could be helpful to end users.
>
> Thank you for your time and feedback,
>
> Authors
>
> [1*] Marsden, A., & Hazan, E. (2025). Universal Sequence Preconditioning. arXiv.
>
>
> .
>
>
> **[SAMPLE SECTION 5.2]**
>
> While Algorithm 2 provides the theoretical foundation for our distillation approach, we developed Algorithm 3 (detailed in Appendix A.4 and A.5) with several practical improvements that significantly enhance performance in applications. Rather than sampling eigenvalues $\alpha$ uniformly in $[0,1]$, Algorithm 3 samples in the range $[-1, 1]$ and deliberately oversamples eigenvalues near $\pm 1$ to better capture high effective memory (see Figure 6), which we use to assemble a large set of pairs $\mathcal{P} = \{(\alpha_i, m_i)\}_i$.
>
> Instead of using a fixed random sample of $h$ eigenvalues, we employ a greedy selection strategy. We start with an initial subset of size $h_{\text{start}}$ chosen to minimize the reconstruction error $\| M^\dagger_{\text{sub}} \Psi_{\text{sub}} - \Phi_{1:k}\|^2_F$, where $\Psi_{\text{sub}}$ is a concatenation of the LDS impulse responses $\mu_{L}(\alpha_i)$ of that subset and $\Phi_{\text{sub}}$ contains the corresponding STU weights, where this subset is chosen through sampling over $\mathcal{P}$. We then incrementally add $h - h_{\text{start}}$ pairs $(\alpha_i, m_i) \in \mathcal{P}$ that most reduce the reconstruction error. After obtaining the initial transformation matrix $\tilde{M}$, Algorithm 3 refines it using gradient descent on the least-squares objective $\|\Phi_{1:k} - \tilde{M}\Psi_{\text{sub}}\|^2_F$, which typically yields approximately $1.4\times$ improvement in reconstruction accuracy.

---

> > ### Comment · Reviewer_ssg6 · 2025-08-05
> >
> > I thank the authors for their extensive reply that addresses my concerns. In particular, I appreciate the motivation to make work as clear and replicable as possible and hope that author's will incorporate all suggestions in their revision. I rise my score accordingly.

---

### Official Review · Reviewer_ZCK4 · 2025-07-03

**Clarity:** 3
**Significance:** 3
**Originality:** 4
**Rating:** 5
**Confidence:** 3

**Summary:**

The paper builds off the work of agarwal et al.’s Spectral Transform Unit. The paper identifies the problem that there is now way to convert a trained STU layer into an explicitly LDS layer. The paper identifies this as being important as the industry seeks a new efficient way of handling particularly long sequences. The paper presents a novel technique for converting STU filters into LDS form. The paper provides the “first provable method to directly learn the parameters of a symmetric LDS of arbitrarily high effective memory and with bounded noise”. The paper’s main result is in algorithm 2, which then offers performance guarantees and empirical verifications for.

**Questions:**

1.Why does epoched future fill in figure three only go until 131072 tokens? (decrease)
2.Can you provide some analysis in the h-vs-accuracy trade off? (increase)
3. Is there a clear way or some promising directions to be able to achieve similar results without the symmetric assumption? (Neutral)

**Ethical Concerns:**

["NO or VERY MINOR ethics concerns only"]

**Final Justification:**

The Authors have adequately addressed my concerns. While my concern regarding Q3 prevents me from increasing my score past a 5 (IE I think the limitation of symmetrical systems is still too great) I do still regard this as a technically solid paper. Thus I will not be changing my score.

**Limitations:**

yes

**Quality:**

4

**Strengths And Weaknesses:**

Quality: Good theory paper with a nice result, theorem 1 wasn’t terribly difficult to follow. Would like more analysis of pseudo-inverse conditions.

Clarity: Figure 1 is barely readable, please make the ‘keys’ better so those with poor eye sight are able to better distinguish the lines. Other than that well organized

Significance: Feels quite significant. The biggest issue is that real-world systems may require non-symmetric dynamics.

Originality: Explicitly novel algorithm and work to go along-side it.

---

> ### Author Rebuttal · Authors · 2025-07-30
>
> Dear Reviewer,
>
> Thank you for your time and constructive feedback on our paper. We appreciate your positive assessment of the theoretical contributions and novelty of our work. We address each point below:
>
>
> **[Pseudo-inverse Analysis]:** We agree that more analysis of the pseudo-inverse conditions would strengthen the paper. In Theorem 1, we show that the error bound depends on $\lambda_{max}$, the largest eigenvalue of the Penrose-Moore pseudo inverse of matrix M. While for $h \ll k$, this can be exponentially large in k, we provide experimental evidence in Appendix A.2 showing that as h grows, $\lambda_{max}$ decays doubly exponentially.
>
>
> **[Figure 1 Readability]:** We will improve Figure 1's accessibility by using more line styles (dashed, dotted, solid) and colors, increasing line thickness, and improving the legend font.
>
> **[Q1 - Epoched Future Fill Memory Limitation]:** The Epoched Future Fill algorithm encounters a CUDA Out of Memory error on our H100-80GB setup at 262,144 tokens. While the theoretical memory complexity is reasonable, the implementation requires intermediate buffers for FFT operations and caches. The naive convolution approaches this limit but remains within bounds, whereas EFF's additional caching and overhead pushes it beyond the threshold. In contrast, SpectraLDS maintains constant memory requirements per token (O(h)) regardless of sequence length.
>
> **[Q2 - h vs. Accuracy Trade-off]:** We provide detailed analysis of this trade-off in Figure 9 (Appendix A.10), which shows reconstruction error as a function of LDS state dimension. For our language modeling experiments, benchmark accuracy remains nearly identical for $h \geq 60$ but degrades rapidly below this threshold. We will expand Table 2 to include performance comparisons at various state dimensions to better illustrate this trade-off.
>
> **[Q3 - Beyond Symmetric Systems]:** We are encouraged by recent developments in this direction. Recently, there has been promising work [1*] extending STUs to broader classes of transition matrices, with error bounds dependent on the maximum magnitude of the imaginary components of eigenvalues. We expect our distillation techniques to apply to these expanded filter sets. In our discussion section, we will add additional discussion in this direction.
>
>
> We believe these revisions will address your concerns while improving the quality of the paper. We would be more than happy to elaborate on any additional concerns you have with the paper, and we appreciate your time and effort.
>
> Thank you,
>
> Authors
>
> [1*] Marsden, A., & Hazan, E. (2025). Universal Sequence Preconditioning. arXiv.

---

> > ### Comment · Reviewer_ZCK4 · 2025-08-05
> > **Comment**
> >
> > The Authors have adequately addressed my concerns. While my concern regarding Q3 prevents me from increasing my score past a 5 (IE I think the limitation of symmetrical systems is still too great) I do still regard this as a technically solid paper. Thus I will not be changing my score.

---

### Official Review · Reviewer_kMfJ · 2025-07-03

**Clarity:** 2
**Significance:** 4
**Originality:** 3
**Rating:** 4
**Confidence:** 3

**Summary:**

This paper proposes SpectraLDS, a provably correct method for distilling Spectral Transform Units (STUs) into linear dynamical systems (LDSs). Since training LDSs directly is often challenging, the authors propose to train the computationally heavier STU instead and then distill it into an LDS to reduce inference cost. Concretely, the goal is to recover LDS parameters $A, B, C$ from the trained STU such that the output of the resulting LDS closely approximates that of the original STU. Theorem 1 shows that the required hidden dimension after distillation is $O(\log L)$, and that the per-token inference cost is reduced from $O(\sqrt{L})$ to $O(\log L)$.

**Questions:**

**(Q1)** In line 54, it is stated that the per-token computation cost is reduced to $O(\log L)$. Does this account for the increased hidden dimension? This seems inconsistent with the explanations in lines 47 and 66, making it difficult to interpret.

**(Q2)** Is the $M_k$ that appears in line 172 different from the matrix $M$ in Algorithm 2? Is my understanding correct that the former corresponds to spectral filters and the latter to LDS filters?

**(Q3)** In line 232, what does the statement "A direct consequence of our approach is that we can distill any high-dimensional symmetric LDS into a low-dimensional LDS with bounded error" mean? Does it imply that the STU corresponds to a high-dimensional LDS?

**(Q4)** In line 254, the phrase “greedily adding 1D-LDS increments” is used. Does this mean that the matrix $\tilde{M}$ is constructed by incrementally adding the vectors $m_i$ obtained from Algorithm 2 one by one?

**(Q5)** Figure 2 shows a visual comparison of filters before and after distillation. Can you also report a quantitative error metric?

**(Q6)** Does the proposed distillation algorithm require access to new data, or is the trained model alone sufficient? Also, how much computational time is needed to run the algorithm?

**Ethical Concerns:**

["NO or VERY MINOR ethics concerns only"]

**Final Justification:**

This paper introduces SpectraLDS, a theoretically grounded method that provably distills Spectral Transform Units (STUs) into linear dynamical systems (LDSs). While STUs are computationally heavier but easier to train, LDSs offer low inference cost yet are harder to train. The proposed distillation is therefore practically important. The paper presents not only Algorithm 2, which has a strong theoretical foundation, but also a more practice-oriented Algorithm 3.

Although the contribution is important and potentially impactful, the submitted version lacks sufficient explanation of Algorithm 3 and of the experiments (especially Figure 1) which hurts clarity. The authors addressed these issues in the rebuttal; accordingly, I assign a borderline accept rating.

**Limitations:**

yes

**Paper Formatting Concerns:**

The authors appear to have deleted the guideline section of the NeurIPS paper checklist.

**Quality:**

2

**Strengths And Weaknesses:**

**Strengths:**

**(S1)** The paper provides a clear summary of the computational costs between existing methods such as Attention, STU, and LDS.

**(S2)** The proposed method is grounded in a solid theoretical framework.

**(S3)** The experiments cover multiple aspects, including comparisons between filters before and after the distillation, inference speed, and downstream language task performance.

**Weaknesses:**

**(W1)** The explanation of Algorithm 3 is insufficient. It is only referenced in lines 224 and 250, and there is almost no description of what the algorithm actually does.

**(W2)** Similarly, Figure 1 is shown in the paper with very little explanation or commentary in the main text.

**(W3)** The numerical experiments are limited to a model size of 340M, which is relatively small in scale.

---

> ### Author Rebuttal · Authors · 2025-07-30
>
> Dear Reviewer,
>
> Thank you for your time and for your feedback, and we appreciate your recognition of the significance of our work. We address each point below:
>
> As Algorithm 3 has many moving parts, we felt that a full explanation of it in the main paper would be confusing to the reader, so we moved the algorithm to appendix A.4 and a full explanation to section A.5.  Based on your feedback, we believe the best approach is somewhere in the middle, where we will instead move section 5.2 to the appendix and introduce a new section 5.2 where we explain how algorithm 3 differs from the main algorithm. We included a sample of what that section may look like below at the end of our response.
>
> For Figure 1, we have a comprehensive description of the experiments performed in Appendix A.7, and in our forthcoming open-source release, we will include all the setup necessary to replicate this experiment. We will move more of the core details from Appendix A.7 to the figure caption and main text so the paper is easier to go through.
>
> Although 340M is not large in scale, for custom model architectures, such as the FlashSTU, evaluating distillation techniques on models of this size is generally common practice (I.e., see [26], which evaluates performance after Laughing Hyena Distillery on models of size 153M and 355M). We tested our technique on all public pretrained FlashSTU [26] models, which included earlier checkpoints of a 550M parameter model as well. We achieved successful distillation without performance drop on that model and have included those results here:
>
> | Model                | MMLU  | Hella. | PIQA  | BoolQ | Wino. | CSQA  | OBQA  | ARC-e | ARC-c | Average |
> |----------------------|-------|--------|-------|-------|-------|-------|-------|--------|--------|---------|
> | Flash STU 550M       | 22.99 | 26.51  | 58.00 | 39.30 | 52.01 | 19.49 | 29.80 | 39.10  | 23.12  | 34.48   |
> | SpectraLDS           | 22.98 | 26.55  | 58.22 | 39.33 | 52.01 | 19.49 | 29.40 | 39.02  | 23.04  | 34.34   |
> | Flash STU Std. Err.  | 0.35  | 0.44   | 1.15  | 0.85  | 1.40  | 1.13  | 2.05  | 1.00   | 1.23   | --      |
> | SpectraLDS Std. Err. | 0.35  | 0.44   | 1.15  | 0.85  | 1.40  | 1.13  | 2.04  | 1.00   | 1.23   | --      |
>
>
>
>
> However as this base model had worse evaluations, we only included our distillation on the 340M model in the final paper. The 550M model was published on 1/28/25, and the 340M model on 4/29/25, so we suspect the greater performance is due to better training choices.
>
> Notably, our technique has shown effectiveness on a large range of evaluations, including modeling large linear dynamical systems with noise and with high effective memory, which we include in Appendix A.11. Although we agree with the reviewer that testing on larger FlashSTU language models would be ideal, it is not feasible for us to train larger FlashSTU language models, and we have tested distillation, and shown strong performance, to the maximal extent we are able to.
>
> **(Q1):** Thank you for this observation. Line 66 is inaccurate and per-token generation costs are O(log L), and we will correct this in the revision.
>
> **(Q2):** We agree this is confusing. The connection is that $[M_1, M_2, … M_K]$ from line 172 is equal to $m_i$ from algorithm 2, where the $M_k$’s were distilled from the LDS defined by the convolutional kernel $\mu_L(\alpha_i)$. In our revision, we will explicitly state that connection in Algorithm 1 and switch Algorithm 2 to refer to the matrix as Q.
>
> **(Q3):** Yes, we can use the STU to learn a given LDS S and we can then distill the STU into another LDS S’. As the STU can learn S with error $\epsilon$ with $k = O(\log L \log \left( \frac{1}{\epsilon} \right))$ filters from line 190, where $L$ is the maximum sequence length, we thus have that we can achieve an $\epsilon$ approximation S’ of S with $O(d_{in} \cdot d_{out} \cdot \log L \cdot \log\frac{1}{\epsilon})$ parameters. Notably, S’ has size independent of the state-dimension of S, and thus for a high-dimensional LDS S, we can distill it into a relatively low-dimensional LDS S’ with bounded error. We agree that the term “high-dimensional” was not clear in this case, and we will clean up this exposition to make clear that our method for LDS-to-LDS distillation has $O(1)$ dependence on the state-dimension of the initial LDS.
>
> **(Q4):** Yes precisely. For Algorithm 3, the matrix M is constructed in part by greedily choosing pairs $(\alpha_i, m_i)$ from a precomputed set so that $|| \Phi_{1:k} - M^{\dagger}\mu_L(\alpha_1, \dots, \alpha_i) || $ is minimal. We will include this description in the revised section 5.2, where we discuss the differences between the improved but more complicated Algorithm 3, and the main algorithm we present.
>
> **(Q5):** For the filters in figure 2, these are the same filters as in Appendix A.8 and A.9, and have reconstruction error of $1.23 \times 10^{-12}$.  For a sense of scale, we show the relationship between LDS dimension and filter reconstruction in Appendix Figure 9. We will add the error metric to the figure caption.
>
> **(Q6):** The proposed distillation algorithm does not require access to new data. In fact, once we have fit the spectral filters with Algorithm 2 or 3, we can immediately transfer the weights of any STU model that uses those filters to a SpectraLDS model. With the 340M FlashSTU model, this process was completed within 150ms. As all FlashSTU models are trained with the same set of 24 spectral filters, we can perform this immediate parameter transfer for any model in the FlashSTU series. The distillation algorithm provides an LDS of a given state dimension per set of spectral filters, and so generally will be rerun only when a different state-dimension is desired, either for more accurate reconstruction or for faster inference. However, it is also worth noting that after the initial step of collecting a set of pairs $(\alpha_i, m_i)$, the remainder of both Algorithm 2 and Algorithm 3 can be computed speedily. In our open-source release, we will provide the set of 61,784 pairs we have computed.
>
> Thank you for your detailed feedback, and we will work on improving the clarity of the paper in the ways you suggested and fixing the minor mistakes. We hope that these fixes largely address your concerns, and we would be happy for you to elaborate on any concerns that you believe still remain in the paper. We also are happy to note that recent work [1*] has extended the STU to linear dynamical systems with asymmetric transition matrices, with bounds based on the maximum imaginary component of the eigenvalues, so we are enthusiastic about further applications of our distillation beyond symmetric LDSs.
>
> Thank you for your time and feedback,
>
> Authors
>
> [1*] Marsden, A., & Hazan, E. (2025). Universal Sequence Preconditioning. arXiv.
>
>
> SAMPLE SECTION 5.2
>
> While Algorithm 2 provides the theoretical foundation for our distillation approach, we developed Algorithm 3 (detailed in Appendix A.4 and A.5) with several practical improvements that significantly enhance performance in applications. Rather than sampling eigenvalues $\alpha$ uniformly in $[0,1]$, Algorithm 3 samples in the range $[-1, 1]$ and deliberately oversamples eigenvalues near $\pm 1$ to better capture high effective memory (see Figure 6), which we use to assemble a large set of pairs $\mathcal{P} = \{(\alpha_i, m_i)\}_i$.
>
> Instead of using a fixed random sample of $h$ eigenvalues, we employ a greedy selection strategy. We start with an initial subset of size $h_{\text{start}}$ chosen to minimize the reconstruction error $\| M^\dagger_{\text{sub}} \Psi_{\text{sub}} - \Phi_{1:k}\|^2_F$, where $\Psi_{\text{sub}}$ is a concatenation of the LDS impulse responses $\mu_{L}(\alpha_i)$ of that subset and $\Phi_{\text{sub}}$ contains the corresponding STU weights, where this subset is chosen through sampling over $\mathcal{P}$. We then incrementally add $h - h_{\text{start}}$ pairs $(\alpha_i, m_i) \in \mathcal{P}$ that most reduce the reconstruction error. After obtaining the initial transformation matrix $\tilde{M}$, Algorithm 3 refines it using gradient descent on the least-squares objective $\|\Phi_{1:k} - \tilde{M}\Psi_{\text{sub}}\|^2_F$, which typically yields approximately $1.4\times$ improvement in reconstruction accuracy.

---

> > ### Comment · Reviewer_kMfJ · 2025-08-06
> >
> > Thank you to the authors for their detailed response. In particular, I appreciate the clarifications regarding Algorithm 3, Figure 1, and questions (Q1)-(Q6). I expect the paper to be revised in accordance with the rebuttal.
> >
> > While 340M is relatively small for language models, I agree that extension to larger models is not strictly necessary. Given the methodological background and the current experimental results, I believe the effectiveness of the approach is sufficiently demonstrated at this stage.
> >
> > Because parts of the current version are hard to follow, my initial assessment leaned toward rejection, and I still consider the paper borderline overall. However, in light of the additional explanations in the rebuttal and the paper’s technical contributions, I am now leaning toward acceptance. Accordingly, I will raise my rating by one point.

---

### Note · Authors · 2025-08-12

We thank the reviewers for the constructive discussion and the opportunity to clarify our work. We would like to use this section to highlight some of the discussion raised throughout the rebuttal periods.

We are really grateful for the reviewers’ recognition of the significance of our work, and we are working hard to improve the accessibility of SpectraLDS to readers unfamiliar with the STU. We believe our work can be nicely explained within the paradigm of SSM recurrent and convolutional duality, and we are working hard to rewrite sections, remove jargon, and add a glossary. We hope for SpectraLDS to be a strong primitive future research can build on, and to this end, we prepared a full open-source code release along with strong pretrained weights to convert from STU-to-LDS, and config files for reproducibility. To further accessibility, in the revised section 5.2, we highlight techniques from Alg. 3 that provide a practical improvement to our main algorithm. We see the SpectraLDS acting as a drop-in replacement for Conv1D layers in sequential models, providing greatly accelerated inference at the cost of some expressivity.

While the restriction to symmetric LDSs is a fair concern, we hope that we have shown that even within this space, there are meaningful challenges and that our method offers a surprisingly effective solution. Moreover, while our theoretical results focus on symmetric LDSs, we have demonstrated that SpectraLDS performs strongly on asymmetric systems, including high-memory regimes where direct LDS fitting is unstable. We note that such results fall in line with prior work (e.g., S4 [10]), and imposing symmetric and near-symmetric structure on the transition matrix has not prevented other architectures from SoTA attention-free performance.

We also note the implications for scaling: we have now provided a strong bound for how increasing LDS parameters corresponds to reducing error. As a corollary of our main result, we have that, with a bounded input sequence, loss is bounded above by $O(e^{-p})$ when fitting an LDS with SpectraLDS, where p is the number of parameters. To our knowledge, this is the strongest scaling law for an LDS, and we hope knowledge of such bounds will be useful in developing a scaling theory around LDSs and structured SSMs.

We are incredibly grateful for the feedback we have received over the course of the rebuttal period, and we are excited to share our algorithm with the broader community!

Thank you,

Authors

---

### Decision · Program_Chairs · 2025-09-17

**Decision:**

Accept (poster)

**Comment:**

Most of the reviewers recommend acceptance, with one recommendation on borderline rejection. The main holdout argument was that the restriction to symmetric transition matrices greatly limits applicability of the algorithm. This was echoed by two other reviewers who gave a 5 rating. The reviewers did agree that while this was a serious limitation, they also all acknowledged that it would be a valuable contribution. The presented evidence suggests that the method is still applicable to real problems despite this limitation. All reviewers also thought that the core part of the algorithm required more explanation, but that could be achieved in revisions. Therefore, the AC believes that the reasons to accept outweigh the holdout reason to reject, and recommends accepting.